# ROBUST TRAINING OF NEURAL NETWORKS AT ARBITRARY PRECISION AND SPARSITY

**Chengxi Ye** **Grace Chu** **Yanfeng Liu** **Yichi Zhang**
**Lukasz Lew** **Li Zhang** **Mark Sandler** **Andrew Howard**

Google DeepMind
{ycx,cxy,yanfengliu,yichizh,lew,zhl,sandler,howarda}@google.com

## ABSTRACT

The discontinuous operations inherent in quantization and sparsification introduce a long-standing obstacle to backpropagation, particularly in ultra-low precision and sparse regimes. While the community has long viewed quantization as unfriendly to gradient descent due to its lack of smoothness, we pinpoint—for the first time—that the key issue is the absence of a proper gradient path that allows training to learn robustness to quantization noise. The standard Straight-Through Estimator (STE) exacerbates this with its well-understood mismatch: a quantization-aware forward pass but oblivious backward pass, leading to unmanaged error and instability. We solve this by explicitly modeling quantization as additive noise, making the full forward-backward path well-defined without heuristic gradient estimation. As one natural solution, we introduce a denoising dequantization transform derived from a principled ridge regression objective, creating an explicit, corrective gradient path that makes learning robust to the noise STE bypasses. We extend this to sparsification by treating it as a special form of quantization that zeros out small values. Our unified framework trains models at arbitrary precisions and sparsity levels with off-the-shelf recipes, enabling stable A1W1 and sub-1-bit networks where others falter. It yields state-of-the-art results, mapping efficiency frontiers for modern LLMs and providing a theoretically grounded path to hyper-efficient neural networks.

## 1 INTRODUCTION

To deploy complex AI models on resource-constrained devices, techniques such as quantization and sparsification are essential. However, their non-differentiable nature poses a long-standing challenge for gradient-based training. For years, the community has relied on the Straight-Through Estimator (STE) (Bengio et al., 2013; Hubara et al., 2018), a surrogate gradient that has been instrumental in enabling Quantization-Aware Training (QAT). While STE has enabled progress, its use often leads to unpredictable and unstable convergence, particularly in demanding low-precision settings. This training instability is a well-documented obstacle; the estimator's fragility is often concealed in large, over-parameterized models (Wang et al., 2023) but becomes apparent on smaller models that are more sensitive to quantization error, where STE-based methods frequently diverge. Such instability has necessitated numerous heuristic modifications, including additional normalization (Zhang et al., 2022), learning rate adjustments Wang et al., 2023, optimizer changes (Liu et al., 2021a), and fine-tuning (McKinstry et al., 2018). These ad-hoc solutions underscore the urgent need for a more principled, robust training framework. Our work addresses this long-standing challenge with a theoretically grounded solution that does not rely on surrogate gradient estimation, instead deriving well-defined, differentiable gradients from a ridge regression objective.

Rather than layering on more empirical fixes, we propose a new QAT framework that addresses the source of this instability. The core mechanism of STE—approximating the derivative of the rounding function as an identity—creates a well-understood mismatch: the forward pass is affected by quantization error, while the backward pass is not. This lack of an explicit, corrective gradient path for the quantization error causes poor convergence in ultra-low-bit QAT. Our non-straight-through method incorporates quantization error into the backward pass via a novel, data-dependent dequan-

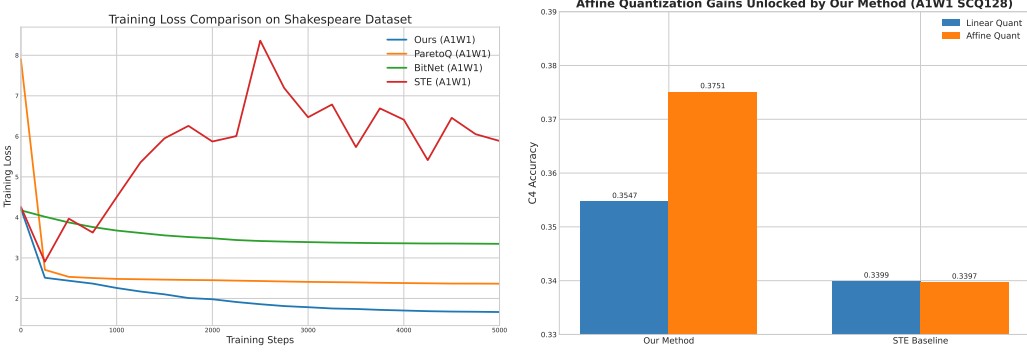

(a) Training Stability on Shakespeare Dataset)   (b) Affine vs. Linear Quantization Performance

Figure 1: **Training Stability and Quantization Robustness Analysis.** (a) Comparison of training loss on the Shakespeare dataset with 1-bit weights and activations (A1W1). Standard STE and BitNet fail to stabilize in this extreme regime, exhibiting divergence or high loss. In contrast, our approach converges smoothly, matching the stability of higher-precision baselines. (b) Comparison of Linear vs. Affine quantization schemes at A1W1. Standard STE (right bars) fails to utilize the additional expressivity of affine parameters, showing no improvement over linear quantization. Our method (left bars) robustly learns the affine parameters (scale and bias), achieving a significant accuracy jump (see Table 1/Appendix A.1.2).

tization step (simple to implement; Code Snippets 1, 2). It provides well-defined gradients without estimation, creating a perturbation-resilient forward pass and error-aware backward gradients.

Our framework's robustness enables a more general approach to training efficient models. Whereas recent state-of-the-art methods often rely on bespoke recipe changes (Wang et al., 2023), architectural modifications, or complex, bit-specific recipe tuning (Liu et al., 2025), our method provides a universal "drop-in" solution that is effective across a wide range of precisions and standard architectures. It also allows us to unlock the full potential of the theoretically superior but rarely used technique of affine quantization. The computational cost of a naive affine implementation has traditionally made it impractical; we overcome this barrier with a novel shortcut formula for affine quantized matrix multiplication that is both robust and efficient (Figure 1(b)).

By treating sparsification as a form of quantization, our unified framework provides a stable foundation to explore the trade-offs between **storage, energy, and quality**. On a Gemma3 1B model (Team et al., 2025), we map the storage-accuracy Pareto frontier (Figure 2), revealing that asymmetric precision (e.g., 4-bit activations 1-bit weights (A4W1)) is optimal for storage. Furthermore, we analyze the trade-off between accuracy and computational efficiency by mapping an approximate energy-accuracy frontier (Figure 3). For this, we use a hardware-agnostic cost metric that estimates the arithmetic computation effort Zhang et al. (2022), a dominant factor in the energy consumption of modern accelerators. This analysis reveals that structured sparsity can simultaneously reduce this approximate computational cost and improve accuracy. These findings reinforce the viability of extremely low-bit precision for high-capacity models, suggesting that aggressive quantization is a key enabler for maximizing model performance within strict hardware constraints. Our key contributions are:

- Pinpointing STE's "quantization-oblivious" backward pass as the key instability source—revealing, for the first time, that proper gradients enable learning to handle quantization noise.

- A simple, robust denoising dequantization transform from ridge regression, enabling—for the first time—stable A1W1 and sub-1-bit training with standard recipes.

- A novel shortcut formula to make affine quantized matrix multiplication computationally efficient, reducing its overhead to a few cheap, low-rank matrix operations.

- State-of-the-art results across a range of challenging, ultra-low precision models, mapping the storage and energy efficiency frontiers for modern LLMs.

## 2 MOTIVATIONS

### 2.1 GHOSTS OF DEPARTED QUANTITIES: THE STE BLIND SPOT

The stability of Quantization-Aware Training (QAT) is critically dependent on its foundational algorithm: the Straight-Through Estimator (STE). For over a decade, STE has been the de facto solution to the non-differentiable nature of quantization, but its reliance on surrogate gradient estimation has perpetuated training instability. A central element of our analysis is reformulating the standard quantization, $y = s \cdot \text{round}\left(\frac{x}{s}\right)$ as an additive perturbation. By defining the rounding error $\delta = \text{round}(\frac{x}{s}) - \frac{x}{s}$, a term that is critically detached and receives no gradient, the forward pass can be expressed as a simple addition: $y = s \cdot (\frac{x}{s} + \delta) = x + s \cdot \delta$.

The issue arises in the backward pass. STE addresses the non-differentiable rounding operation by replacing its true derivative with a surrogate (Hubara et al., 2018; Yin et al., 2019). In the most common case, this surrogate is the identity function, yielding $\frac{dy}{dx} = 1$. The gradient of the loss with respect to $x$ is then computed as $\frac{dL}{dx} = \frac{dL}{dy} \cdot \frac{dy}{dx} = \frac{dL}{dy}$. As is clear from this equation, the quantization error term $\delta$ is **completely absent from the gradient computation**.

In essence, STE creates a system where the forward pass is quantization-aware, but the backward pass is *quantization-oblivious*. This is a "critical blind spot": the quantization error $\delta$, despite affecting the forward pass, vanishes from the gradient, becoming, in the words of Bishop Berkeley's critique of calculus, a "ghost of a departed quantity" Berkeley (1754). Because the update signal is blind to this error, due to the simplistic gradient approximation, preceding layers have no opportunity to learn to handle the perturbation. This unmanaged error corrupts the learning signal, leading to training divergence (Figure 1(a)).

### 2.2 THE AFFINE QUANTIZATION DILEMMA

A robust framework must also handle the asymmetric data distributions common in neural networks, which can degrade performance if the quantization grid is misaligned (e.g., aligning non-negative outputs from a ReLU function to an integer grid of $\{-2, -1, 0, 1\}$ whose mean is negative). The principled solution is the affine quantization $s \cdot q + b$, but it suffers from two major drawbacks directly linked to STE's blind spot. First, STE's error-oblivious gradient struggles to optimize the sensitive bias term $b$, often yielding little to no quality gain (Figure 1(b)). Second, its naive implementation is computationally expensive, undermining the goal of efficiency. A core motivation of our work is to unlock the benefits of affine quantization without these prohibitive costs.

### 2.3 THE PROMISE OF BITWISE COMPUTATION: A1W1 AND BEYOND

Solving these foundational issues facilitates the robust training of networks at extremely low precision, such as 1-bit for activations and weights (A1W1). At this level, expensive floating-point matrix multiplications can be replaced by highly efficient bitwise operations, specifically XNOR and popcount (bit-counting). This capability opens the door to hardware architectures that are significantly simpler, faster, and more power-efficient. Furthermore, while Spiking Neural Networks (SNNs) have long sought to achieve similar efficiency by mimicking discrete biological impulses (Hodgkin & Huxley, 1952; Dayan & Abbott, 2005), their non-differentiable nature poses significant training challenges (Yamazaki et al., 2022; Lee et al., 2016). As this paper demonstrates, our robust training algorithm for 1-bit networks offers a practical alternative, achieving comparable computational sparsity and efficiency within a standard gradient-based framework.

## 3 A THREE-STAGE METHOD FOR ROBUST QUANTIZATION

Our framework conducts quantization-aware training using a three-stage process. This method applies to both quantization and sparsification, which we treat as a special form of quantization that maps insignificant values to zero.

### 3.1 STAGE 1: PREQUANTIZATION TRANSFORM ($f$)

First, we apply a prequantization transform $f$, to map the high-precision input tensor $x$ into a range suitable for integer or low-precision float rounding. For zero-centered data like weights or for easy-to-quantize regimes, we may use a simple linear transform $f(\boldsymbol{x}) = \frac{\boldsymbol{x}}{s_f}$. [1] For activations or in ultra-low-precision regimes where asymmetry is critical (e.g., 2-bit values of $\{-2, -1, 0, 1\}$), we use an affine transform $f(\boldsymbol{x}) = \frac{\boldsymbol{x} - b_f \cdot \mathbf{1}}{s_f}$ to optimally align the data with the quantization grid. [2]

### 3.2 STAGE 2: QUANTIZATION ERROR INJECTION ($\delta$) AND THE STE BLIND SPOT

The second stage of our framework models the quantization step as an additive **quantization error**, $\boldsymbol{\delta}$. The quantized low-precision vector, $\boldsymbol{q}$, is defined as the sum of the scaled transform and this new error term:

$$\boldsymbol{q} = f(\boldsymbol{x}) + \boldsymbol{\delta} \tag{1}$$

Here, $\boldsymbol{\delta} = \text{round}(f(\boldsymbol{x})) - f(\boldsymbol{x})$ represents the error introduced by the rounding operation. This formulation is general: for integer quantization, rounding is to the nearest integer; for low-precision float formats like FP4, it is to the nearest representable float value. Crucially, $\boldsymbol{\delta}$ is intentionally **detached** from the computation graph so that it **receives no gradient** in the backward pass.

This detached error, $\boldsymbol{\delta}$, is the central problem that causes training instability. The standard Straight-Through Estimator (STE) navigates the non-differentiable rounding function by approximating its local derivative as an identity ($\frac{d\boldsymbol{q}}{df(\boldsymbol{x})} = \boldsymbol{I}$) or other surrogates (Bengio et al., 2013; Hubara et al., 2018; Yin et al., 2019). The gradient calculation is completely blind to the error that was introduced, and this unmanaged perturbation corrupts the learning signal that propagates to the preceding layers.

### 3.3 STAGE 3: DEQUANTIZATION WITH A DENOISING TRANSFORM ($g$)

A core innovation of our work lies in the dequantization step which maps the quantized data back to the original floating-point range to approximate the unquantized data. While typical methods simply invert the scaling from Stage 1, our approach introduces a **denoising dequantization transform**, $g$, which is explicitly designed to solve the problem of the unmanaged quantization error, $\boldsymbol{\delta}$. Unlike heuristic gradient estimators, we formulate this dequantization as a principled ridge regression problem, yielding well-defined gradients that ensure numerical stability against low-variance data without any surrogate approximations.

#### 3.3.1 FORMULATION: GENERAL AND SYMMETRIC TRANSFORMS

**General (Affine) Dequantization for Uncentered Data** For uncentered data, such as activations, we use a full affine transform, $g(\boldsymbol{q}) = s_g \cdot \boldsymbol{q} + b_g$. We find the optimal scale $s_g$ and offset $b_g$ by solving the ridge regression objective:

$$\min_{s_g, b_g} \frac{1}{2N} \left\| s_g \cdot \boldsymbol{q} + b_g \cdot \mathbf{1} - \boldsymbol{x} \right\|^2 + \frac{\lambda}{2} s_g^2 \tag{2}$$

where $\lambda$ is a regularization factor. The closed-form solution for the dequantized vector, is:

$$g(\boldsymbol{q}) = \frac{\text{Cov}_{xq}}{\text{Var}_q + \lambda}(\boldsymbol{q} - \overline{\boldsymbol{q}}) + \overline{\boldsymbol{x}} \tag{3}$$

The regularization parameter $\lambda$ acts as a "denoising" knob. As $\lambda \to \infty$, the scale $s_g \to 0$, and the dequantization collapses to the mean: $\overline{\boldsymbol{x}}$, forcing the transform to ignore the "noisy" quantized vector $\boldsymbol{q}$ and fall back to the most stable component of the original signal: its mean. This provides a mechanism to balance signal fidelity against noise suppression, which is critical for stabilizing the backward pass. A single value of $\lambda = 0.01$ proved sufficient to ensure stable training across all our diverse experimental settings.

---

[1] The parameter for the linear quantization is $s_f = \frac{\max(|\boldsymbol{x}|)}{q_{max}}$.

[2] The parameters for the affine case can be computed as $s_f = \frac{x_{\max} - x_{\min}}{q_{\max} - q_{\min}}$, and $b_f = x_{\min} - s_f \cdot q_{\min}$.

**Symmetric (Linear) Dequantization for Centered Data** For data that is naturally centered around zero, the dequantization simplifies to a more efficient **linear transform**, $g(\boldsymbol{q}) = s_g \cdot \boldsymbol{q}$. The optimal scale $s_g$ is found by solving a simplified, bias-free objective:

$$\min_{s_g} \frac{1}{2N} \|s_g \cdot \boldsymbol{q} - \boldsymbol{x}\|^2 + \frac{\lambda}{2} s_g^2 \tag{4}$$

This yields a simpler closed-form solution for the scaling factor: $s_g = \frac{\langle \boldsymbol{q}, \boldsymbol{x} \rangle}{\langle \boldsymbol{q}, \boldsymbol{q} \rangle + \lambda}$.

### 3.3.2 HOW THIS SOLVES THE STE BLIND SPOT

Our method rectifies the information-flow problem in STE through a two-part mechanism:

**1. The Forward Pass: The Error is Included.** The input to the dequantization transform $g$ is not the clean value $f(\boldsymbol{x})$; it is the noisy, quantized vector $\boldsymbol{q} = f(\boldsymbol{x}) + \boldsymbol{\delta}$. This means the quantization error $\boldsymbol{\delta}$ is fundamentally part of the input to the dequantization step.

**2. The Backward Pass: The Gradient is Error-Aware.** During backpropagation, the gradient with respect to the quantized vector q is computed via the chain rule: $\frac{dL}{d\boldsymbol{q}} = \frac{dL}{dg(\boldsymbol{q})} \frac{dg(\boldsymbol{q})}{d\boldsymbol{q}}$. Because the parameters of $g$ (its scale and offset) are calculated from the statistics of $\boldsymbol{q}$ (Eq. 3), its derivative is directly shaped by the values within $\boldsymbol{q}$. Since $\boldsymbol{q}$ contains the error $\boldsymbol{\delta}$, this local derivative $\frac{dg(\boldsymbol{q})}{d\boldsymbol{q}}$ becomes an explicit function of that error (Sec. A.4). By forcing the quantization error to participate in the backward pass, our transform provides the learning signal that STE discards, allowing preceding layers to adapt their weights and become robust to the error.

### 3.4 SPARSIFICATION AS A SPECIAL FORM OF QUANTIZATION

Our framework seamlessly extends to network sparsification by treating it as a special form of quantization that maps only insignificant values to zero. Our framework unifies these techniques by modeling them as sequential, additive error injections.

First, a hard-thresholding operation is applied to the full-precision tensor x to enforce a specific sparsity pattern, such as 2:4 structured sparsity. This non-differentiable step introduces the first source of error, the sparsity error $\boldsymbol{\delta}_S = \text{threshold}(\boldsymbol{x}) - \boldsymbol{x}$. The resulting sparse tensor is $\boldsymbol{x}_S = \boldsymbol{x} + \boldsymbol{\delta}_S$. Next, this sparse tensor $\boldsymbol{x}_S$ is fed into the quantization pipeline described in Section 3.2. This stage introduces the second source of error, the quantization error $\boldsymbol{\delta}_Q$, resulting in the final low-precision, sparse tensor $\boldsymbol{q} = f(\boldsymbol{x}_S) + \boldsymbol{\delta}_Q$.

The power of our unified framework lies in the final dequantization stage. The denoising transform $g(\boldsymbol{q})$ is applied to this doubly-perturbed tensor $\boldsymbol{q}$ with the objective of reconstructing the original, dense, high-precision tensor x. Because the parameters of $g$ are derived from the statistics of $\boldsymbol{q}$, the transform inherently learns to correct for the combined error distribution from both $\boldsymbol{\delta}_S$ and $\boldsymbol{\delta}_Q$. Since the input to the dequantization $\boldsymbol{q} = f(\boldsymbol{x}_S) + \boldsymbol{\delta}_Q$ is the tensor after both sparsity and quantization errors have been injected, the ridge regression objective (Eqs. 2, 4) implicitly minimizes the loss to the unperturbed $\boldsymbol{x}$ with respect to the combined perturbation. The backward pass is aware of the total perturbation, allowing the network to become robust to both compression techniques.

## 4 THE DEQUANTIZATION TRANSFORM AS A NORMALIZATION LAYER

Our framework can be further understood through a powerful analogy: the denoising dequantization transform $g$ also acts as a **normalization layer** applied directly to the noisy, quantized vector $\boldsymbol{q}$. This perspective highlights two complementary benefits: the normalization structure creates a corrective gradient pathway, and the $\lambda$ term provides numerical stability akin to the $\epsilon$ in LayerNorm. This normalization-based view explains both the stability and efficiency of our method.

Comparing our transform to a standard LayerNorm: $\gamma \cdot \frac{\boldsymbol{q} - \mu_q}{\sigma_q} + \beta$, reveals the structural similarity:

- **Our Denoising Transform** $g$: $\underbrace{\left(\dfrac{\mathrm{Cov}_{xq}}{\sqrt{\mathrm{Var}_q + \lambda}}\right)}_{\text{Analogous to } \gamma} \cdot \underbrace{\dfrac{q - \overline{q}}{\sqrt{\mathrm{Var}_q + \lambda}}}_{\text{Analogous to } \frac{q - \mu_q}{\sigma_q}} + \underbrace{\overline{x}}_{\text{Analogous to } \beta}$

This formulation reveals that our transform normalizes the noisy quantized vector $q$ and then rescales and shifts it to best approximate the statistics of the original high-precision vector $x$. The computational cost of these operations is on par with a LayerNorm. When the bias term is not included, the transform simplifies and its complexity falls back to being on par with an RMSNorm.

## 5 EFFICIENT AFFINE QUANTIZED MATRIX MULTIPLICATION

A two-sided affine transform provides the most robust quantization, and applying it on a per-channel basis is critical for preserving quality. A naive implementation of this operation, $\tilde{Y} = \tilde{X} \cdot \tilde{W}$, expands into a sum of four terms, making it complex to implement efficiently. We introduce a novel shortcut that proves this high-quality approach is both elegant and fast.

### 5.1 A NOVEL SHORTCUT ENABLED BY THE $L_2$ FORMULATION

Our method is built upon a mean-centering identity, which decomposes a matrix product into its mean-centered and mean-component interactions:

$$Y = X \cdot W = (X - \overline{x} \cdot \mathbf{1}^T) \cdot (W - \mathbf{1} \cdot \overline{w}^T) + \overline{x} \cdot \overline{w}^T n \tag{5}$$

where $\overline{x}$ and $\overline{w}$ denote the mean vectors, and $n$ represents the inner product dimension size.

This identity provides the key structure to dramatically simplify the affine dequantization process, as formalized in the following theorem.

**Theorem 1.** *The result of a two-sided, channel-wise affine dequantization, $\tilde{Y}$, can be expressed as:*

$$\tilde{Y} = (\boldsymbol{s}_X \cdot \boldsymbol{s}_W^T) \odot (Q^X \cdot Q^W - \overline{\boldsymbol{q}}_X \cdot \overline{\boldsymbol{q}}_W^T n) + \overline{\boldsymbol{x}} \cdot \overline{\boldsymbol{w}}^T n \tag{6}$$

*where $n$ is the inner dimension size, the bar notation $\overline{(\cdot)}$ denotes the mean, variables with $X$ are column vectors (row-wise statistics), and variables with $W$ are column vectors (column-wise statistics) that are transposed where appropriate.*

*Proof.* By substituting equation 3 into the quantized matrix multiplication:

$$\begin{aligned}
\tilde{X}\tilde{W} &= ((\boldsymbol{s}_X \cdot \mathbf{1}^T) \odot (Q^X - \overline{\boldsymbol{q}}_X \cdot \mathbf{1}^T) + \overline{\boldsymbol{x}} \cdot \mathbf{1}^T) \cdot ((Q^W - \mathbf{1} \cdot \overline{\boldsymbol{q}}_W^T) \odot (\mathbf{1} \cdot \boldsymbol{s}_W^T) + \mathbf{1} \cdot \overline{\boldsymbol{w}}^T) \\
&= ((\boldsymbol{s}_X \cdot \mathbf{1}^T) \odot (Q^X - \overline{\boldsymbol{q}}_X \cdot \mathbf{1}^T)) \cdot ((Q^W - \mathbf{1} \cdot \overline{\boldsymbol{q}}_W^T) \odot (\mathbf{1} \cdot \boldsymbol{s}_W^T)) + \overline{\boldsymbol{x}} \cdot \overline{\boldsymbol{w}}^T n \\
&= (\boldsymbol{s}_X \cdot \boldsymbol{s}_W^T) \odot (Q^X Q^W - \overline{\boldsymbol{q}}_X \cdot \overline{\boldsymbol{q}}_W^T n) + \overline{\boldsymbol{x}} \cdot \overline{\boldsymbol{w}}^T n
\end{aligned} \tag{7}$$

$\square$

**Interpreting the Shortcut Formula**  Our theorem's efficiency comes from recasting a complex expansion into a simple structure: a standard linear term plus two cheap, rank-1 corrections. The main term, $(\boldsymbol{s}_X \cdot \boldsymbol{s}_W^T) \odot (Q^X \cdot Q^W)$ **is the standard computation for linearly quantized matrix multiplication**. This is supplemented by two offset corrections: a novel subtraction term that centers the product based on the means of the quantized data $-\overline{\boldsymbol{q}}_X \cdot \overline{\boldsymbol{q}}_X^T n$, and an addition that reconstructs the output's correct mean using the original high-precision data $\overline{\boldsymbol{x}} \cdot \overline{\boldsymbol{w}}^T n$. This efficient structure proves that robust, channel-wise affine dequantization is nearly as fast as standard linear quantization, reducing the computational overhead from four matrix terms to a single integer matrix multiplication and two low-rank corrections (Code Snippet 3).

## 6 EXPERIMENTS

We conduct a comprehensive set of experiments to validate the robustness and efficiency of our quantization-aware training framework. Our evaluation spans a range of model scales and architectures, from small-scale transformers to state-of-the-art Gemma LLMs (Team et al., 2025). A key

finding is that while many standard quantization methods are benchmarked on large, overparameterized models where their flaws can be masked, these methods often falter on smaller models that have less redundancy. Our framework, in contrast, demonstrates superior stability and performance across all model scales.

While this section focuses on our primary results on modern transformer architectures, the appendix provides a comprehensive set of supplementary experiments and analyses. These materials offer a deeper validation of our framework's components. We analyze the consistent superiority of our method over the standard STE (Sec. A.1.1), quantify the benefits of affine quantization unlocked by our approach (Sec. A.1.2), and explore the application of advanced techniques, including our centered Hadamard transform (Sec A.1.3), low-precision FP4 formats (Sec. A.1.4), and the trade-offs in structured sparsity (Sec. A.1.5). Furthermore, to demonstrate the broad applicability of our method, the appendix presents results on established benchmarks beyond large-scale language modeling, including ResNet-50 on ImageNet and Transformers on WMT machine translation tasks, providing a comprehensive validation of our framework's robustness and versatility.

## 6.1 Experimental Setup

To test robustness, hyperparameters matched BF16 baselines. Our formulation eliminates need for tuned schedules. All use channel-wise or sub-channel quantization (SCQ, block 128) for fine-grained quantization. Notation: "AxWy" for x-bit activations, y-bit weights. For our quantization-only experiments (1, 2, and 4-bit), we use affine quantization to effectively model asymmetric data distributions. In contrast, for all experiments involving ternary quantization (1.5-bit) or structured sparsity, we intentionally use the simpler linear quantization, which naturally produces representations favorable for hardware simplicity. The implementation frameworks are tailored to the experiment scale: small-scale experiments on nanoGPT models are conducted in PyTorch on NVIDIA A100 GPUs, whereas large-scale pre-training on Gemma LLMs is implemented in JAX and trained on up to 64 TPUs.

## 6.2 Core Method Validation on nanoGPT

We begin by validating our method's stability on small-scale GPT models from the nanoGPT repository (Karpathy, 2022). This low-resource setting effectively highlights the fragility of standard methods. We use the Shakespeare dataset, consisting of approximately one million characters, to train a transformer model with 6 layers, 6 attention heads, and 11 million parameters. We apply channel-wise quantization on the model. As shown in Figure 1(a), our method, using its robust affine quantization, converges smoothly even when both activations and weights are quantized to 1-bit (A1W1). In a direct comparison, the affine version of STE exhibits significant instability at the same precision—an issue that could not be resolved by simply reducing the learning rate. These results validate our theoretically grounded solution, which succeeds without gradient estimation where long-standing methods fail. For baseline comparisons, we utilize the publicly available symmetric implementations of BitNet (Wang et al., 2023) and ParetoQ (Liu et al., 2025), as they lack native support for affine quantization. Our results indicate that our method's inherent stability enables it to effectively leverage the superior modeling capacity of affine quantization, a regime where other methods are either demonstrably unstable or lack native support.

To validate these findings on a more realistic task, we trained a GPT-2 small model (124M parameters) on the OpenWebText dataset for 25k steps. The results, shown in Figure 8, confirm our initial findings. Our method again demonstrates highly stable training, while both BitNet and STE show unstable training, with their validation losses becoming erratic or resulting in NaNs. While ParetoQ also trains, it converges to a significantly worse loss.

## 6.3 Scaling to State-of-the-Art LLMs: A Gemma 1B Case Study

To validate that our framework scales effectively to modern architectures, we performed a comprehensive analysis on the Gemma 1B model. We conducted a miniature pre-training run, training the model for 25k steps on 6.5B tokens with a context length of 512 from the C4 dataset. This setup serves as a rigorous testbed to map the Pareto frontiers for storage and computational efficiency, allowing us to identify the optimal strategies for quantizing state-of-the-art language models.

### 6.3.1 The Storage Efficiency Frontier: Asymmetric Quantization is Key

Our first analysis examines the trade-off between model accuracy and the storage cost of the quantized weights, measured in effective bits per element (BPE). As illustrated in Figure 2, our method maps out a clear and compelling Pareto frontier, demonstrating a practical path to extreme model compression without significant performance degradation.

The central finding is that the optimal storage frontier is achieved not through a symmetric quantization scheme (e.g., A2W2), but through a distinctly **asymmetric strategy**: pairing robust 4-bit activations (A4) with aggressively quantized, ultra-low-bit weights (e.g., W1). This approach is effective because it allocates the precision budget intelligently, preserving activation precision to maintain information flow while leveraging the static nature of weights for extreme compression. By integrating this principle with structured sparsity, our framework can push weights into the sub-1-bit regime while preserving the model's core capabilities.

A central dilemma in quantization is how to handle outliers. While some methods clip these values, this risks discarding valuable information. A more recent approach, the Hadamard transform (Tseng et al., 2024; Ashkboos et al., 2024; Liu et al., 2024; Panferov et al., 2025), acknowledges their potential value by blending outliers into other coordinate axes.

Our framework proposes a third, more direct approach. We also acknowledge outliers may be features and, like some coordinate rotation methods, we "blend" them; however, we blend their influence into the computation of the statistical dequantization parameters (Eq. 3) rather than the coordinates. Both outliers and normal samples participate in this computation, ensuring all samples contribute to the information flow. This is made robust through subchannel quantization (SCQ), which localizes this statistical blending. An outlier's corrupting influence is contained within its own moderate-sized block, allowing its information to be preserved without distorting the quantization for the rest of the channel. Our results confirm that this strategy—local statistical blending via SCQ—defines a superior Pareto frontier, establishing a more direct and practical path to state-of-the-art storage efficiency than methods relying on complex transforms like Hadamard.

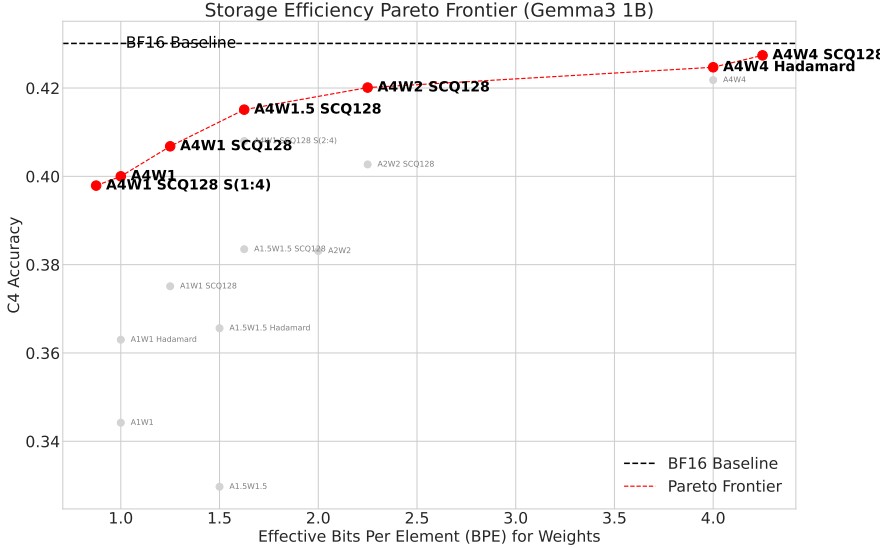

Figure 2: **Storage Efficiency Frontiers:** We map the trade-off between validation accuracy and effective bits-per-element (BPE). The frontier reveals that asymmetric quantization (e.g., A4W1) provides a superior storage-accuracy trade-off compared to symmetric settings (e.g., A2W2), as it preserves activation information while aggressively compressing static weights.

### 6.3.2 The Energy Efficiency Frontier: The Synergistic Power of Sparsity

The energy efficiency analysis, shown in Figure 3, reinforces these findings and reveals a powerful synergy with structured sparsity. We use a hardware-agnostic proxy for arithmetic energy cost Zhang et al. (2022), defined as (Sparsity Factor) × (Activation Bits) × (Weight Bits) × (Total Operations). It is important to note that this metric is a first-order approximation and deliberately omits the dom-

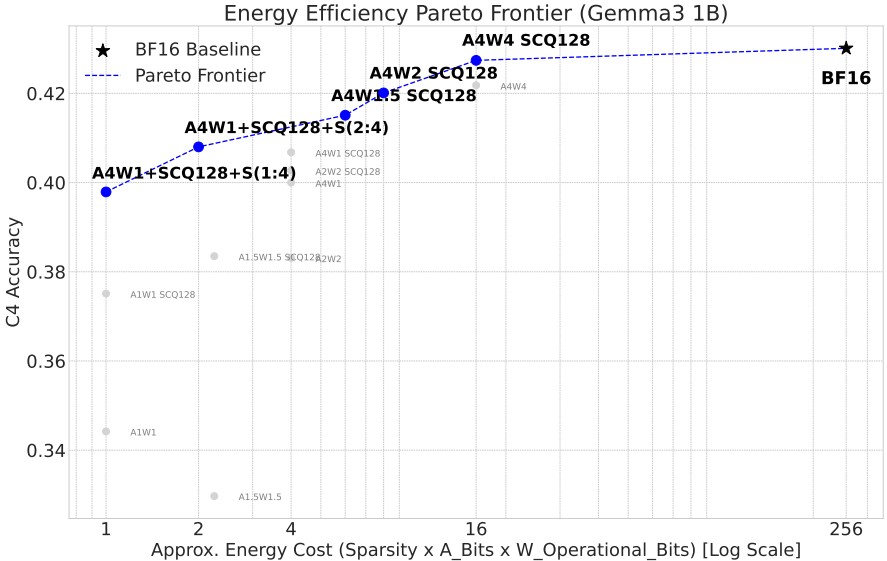

Figure 3: **Approximate Energy Efficiency Frontier:** Estimating compute cost (Activation Bits × Weight Bits × Sparsity). The results demonstrate a synergy between our estimator and structured sparsity: the (1:4 and 2:4) structured sparse A4W1 models reduce the compute cost of the dense equivalent while maintaining high accuracy, defining the efficiency frontier.

inant costs of data movement (Neseem et al., 2024), quantization overhead, or other non-arithmetic operations, but serves as a principled tool for comparing the theoretical arithmetic efficiency of different bit allocations. This score represents a principled lower bound on the true energy cost.

Our analysis identifies two critical results: First, for any given computational budget, an asymmetric bit allocation that prioritizes activation precision remains the superior strategy. Second, and most notably, introducing 2:4 sparsity to the A4W1 model simultaneously cuts the computational cost in half while also increasing model accuracy (from 0.4068 to 0.4080). This outcome confirms that the most effective path to efficiency is through the synergistic combination of our core denoising transform, asymmetric bit allocation, sub-channel quantization, and structured sparsity.

## 6.4 Scaling to Larger Models: A 4B Parameter Case Study

To investigate whether a larger model, when quantized to a similar footprint, can outperform a smaller model, we conducted a 26B token pre-training run comparing the 700M-parameter Gemma3 1B against the 3.2B-parameter Gemma3 4B.

### 6.4.1 Storage and Compute Efficiency at Scale

Our results, visualized in Figure 4, demonstrate that a larger model aggressively quantized with our method can be superior to a smaller model, even one at higher precision. The Gemma3 4B model, quantized to A4W1 with 2:4 sparsity, achieves higher accuracy (0.4517) than both the BF16 Gemma3 1B (0.4494) and the quantized A4W4 Gemma3 1B (0.4443). The energy efficiency analysis reveals an even clearer advantage: the sparse, quantized 4B model is not only more accurate than a quantized 1B model but achieves this with a significantly lower total computational cost. For a detailed tabular comparison across all 1B and 4B configurations, please refer to Table 6 in Appendix A.1.6.

### 6.4.2 Scaling Properties and Hardware Implications

These experiments confirm that for a fixed budget, aggressively quantizing a larger model yields superior performance compared to a smaller, high-precision model. This trade-off offers distinct advantages for hardware implementation. Ultra-low-precision models, particularly those operating at the 1-bit or ternary level, can be executed on significantly simplified hardware architectures.

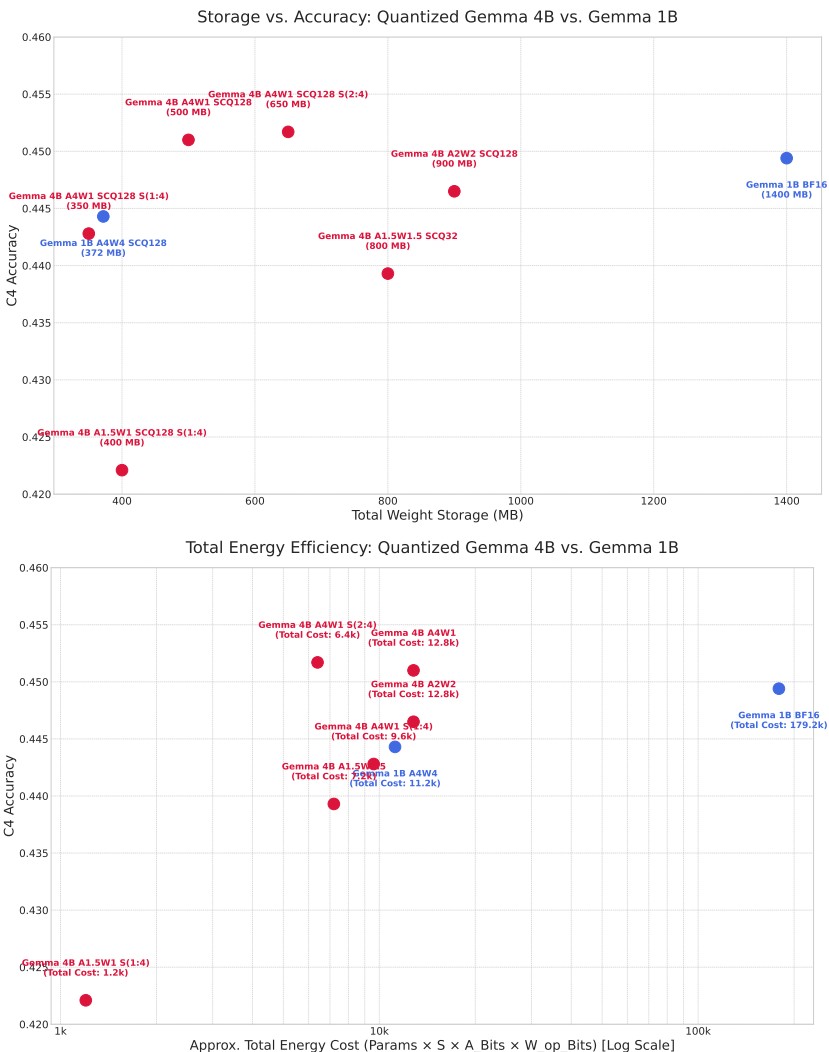

Figure 4: (a) Storage vs. Accuracy comparison between Gemma3 1B and 4B models. The quantized 4B model achieves higher accuracy than both BF16 and quantized versions of the 1B model. (b) Total Energy Cost vs. Accuracy. The quantized and sparse 4B model is both more accurate and more computationally efficient than a quantized 1B model.

By replacing complex floating-point units with efficient bitwise operation circuits, we can achieve substantial reductions in power consumption, silicon area, and manufacturing costs. This efficiency paves the way for deploying high-capacity models on edge devices and specialized accelerators where energy constraints are paramount.

## 7 SUMMARY

The non-differentiable nature of quantization has long challenged efficient model training, as the community has viewed it as inherently unfriendly to gradient descent's reliance on smoothness. We pinpoint—for the first time—that the core issue is the lack of a proper gradient path allowing training to learn robustness to quantization noise, which STE exacerbates by ignoring error in the backward pass. Our work provides a theoretically sound resolution through a ridge regression-derived denoising dequantization transform, ensuring well-defined gradients without heuristic estimation. This unified framework enables robust training at arbitrary precisions and sparsity with standard recipes and no ad-hoc fixes, unlocking the full potential of low-precision computing for resource-constrained applications.

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

# A APPENDIX

This appendix presents supplementary experiments and analyses that further validate our framework's robustness and versatility. We begin with detailed ablation studies: we analyze our method's consistent superiority over standard STE (Sec. A.1.1), quantify the gains unlocked by affine quantization (Sec. A.1.2), and explore advanced techniques including our centered Hadamard transform (Sec. A.1.3), low-precision FP4 formats (Sec. A.1.4), and structured sparsity trade-offs (Sec. A.1.5). Additionally, we outline our energy consumption methodology in Sec. A.1.6. To demonstrate applicability beyond LLM pre-training, we present results on ResNet-50 (Sec. A.1.7) and WMT machine translation tasks (Sec. A.1.8).

On the implementation side, we detail the computational flow of quantized matrix multiplication in Sec. A.2, analyze the complexity of the affine shortcut in Sec. A.3 and provide a rigorous gradient analysis in Sec. A.4. Finally, we provide an extended discussion of related work in Sec. A.5 and include reference code in Sec. A.6.

## A.1 ABLATION STUDIES AND ANALYSIS

### A.1.1 OUR METHOD CONSISTENTLY OUTPERFORMS STE

Across all tested configurations, our **denoising reconstruction** method consistently outperforms the standard **Straight-Through Estimator (STE)**. While the improvement is present at all bit-widths, the performance gap widens significantly at lower precisions, highlighting our method's superior stability and accuracy in the most challenging regimes (Table 1, Figure 5).

- At a higher precision like **A4W4** (Affine, SCQ 128), our method achieves an accuracy of **0.4274** compared to STE's **0.4254**.

- The advantage grows at **A2W2** (Affine, SCQ 128), where our method scores **0.4027** while STE only reaches **0.3794**.

- Most critically, in the **A1.5W1.5** channel-wise setting, STE fails to converge, whereas our method remains stable and achieves a respectable accuracy of **0.3297**, demonstrating its fundamental robustness.

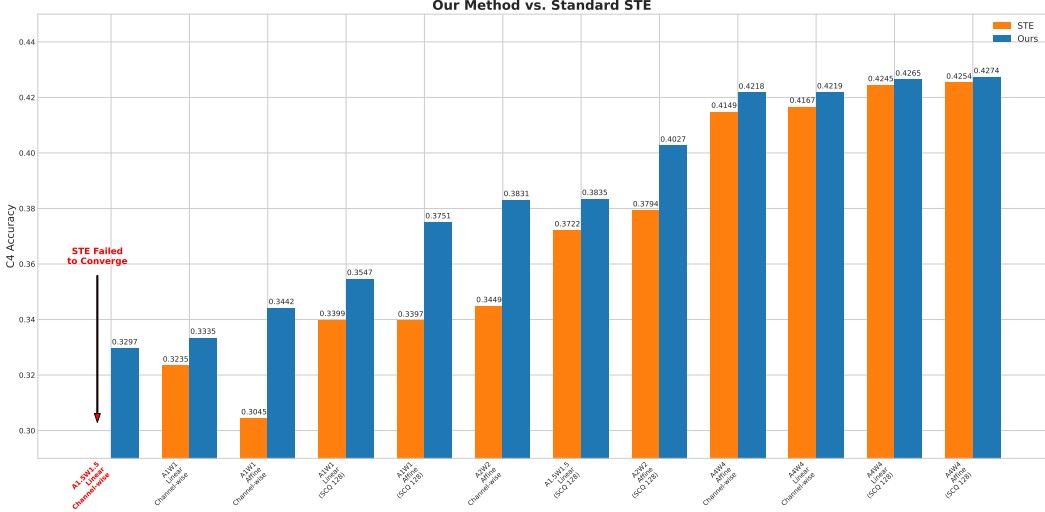

Figure 5: Comparison of our denoising reconstruction method against STE across all experiment configurations, sorted by our method's performance. Our approach consistently yields higher accuracy, and the improvement is most pronounced at lower bit-widths. Notably, in the A1.5W1.5 channel-wise setting, STE fails to converge entirely.

Table 1: Comparison of our denoising reconstruction method against STE across various configurations. Our method consistently yields higher accuracy, with the most significant gains at lower bit-widths. The STE run for A1.5W1.5 channel-wise Linear quantization failed to converge.

| Configuration | Quantization | STE Acc. | Our Method Acc. | Improvement |
|---|---|---|---|---|
| A1W1 | Linear, Channel | 0.3235 | 0.3335 | +0.0100 |
| A1W1 | Affine, Channel | 0.3045 | 0.3442 | +0.0397 |
| A1W1 | Linear, SCQ 128 | 0.3399 | 0.3547 | +0.0148 |
| A1W1 | Affine, SCQ 128 | 0.3397 | 0.3751 | +0.0354 |
| A1.5W1.5 | Linear, Channel | NaN | 0.3297 | - |
| A1.5W1.5 | Linear, SCQ 128 | 0.3722 | 0.3835 | +0.0113 |
| A2W2 | Affine, Channel | 0.3449 | 0.3831 | +0.0382 |
| A2W2 | Affine, SCQ 128 | 0.3794 | 0.4027 | +0.0233 |
| A4W4 | Linear, Channel | 0.4167 | 0.4219 | +0.0052 |
| A4W4 | Affine, Channel | 0.4149 | 0.4218 | +0.0069 |
| A4W4 | Linear, SCQ 128 | 0.4245 | 0.4265 | +0.0020 |
| A4W4 | Affine, SCQ 128 | 0.4254 | 0.4274 | +0.0020 |

### A.1.2 AFFINE QUANTIZATION GAINS ARE UNLOCKED BY OUR METHOD

Our results show that while affine quantization is theoretically superior for handling asymmetric data, **only our method can reliably unlock its benefits, especially at low precision**. In contrast, STE often fails to gain any advantage from an affine transform and can sometimes even perform worse (Table 2, Figure 6).

With our method at **A1W1** (SCQ 128), **Affine (0.3751)** significantly outperforms **Linear (0.3547)**, demonstrating a clear benefit. However, with the standard STE baseline at **A1W1** (channel-wise), **Affine STE (0.3045)** is surprisingly *worse* than **Linear STE (0.3235)**.

This supports our conjecture that the learnable bias term in affine quantization is highly sensitive to outliers. STE's "quantization-oblivious" gradient cannot optimize this sensitive parameter effectively, leading to a misaligned quantization grid. The ability of our framework to correctly handle this error is therefore a critical advantage.

Conversely, at higher precisions such as 4-bit, the benefits of affine quantization diminish significantly. Our A4W4 experiments show that a simple linear transform achieves nearly identical results to the more complex affine version (e.g., **0.4265** for Linear with our method vs. **0.4274** for Affine with our method, both with SCQ 128). This suggests that while correctly modeling asymmetries is critical in ultra-low-bit regimes, its importance is less pronounced when more bits are available.

Table 2: Comparison of quantization schemes. For each configuration, the highest accuracy for each method (Our Method vs. STE) is in bold. Our method consistently benefits from a full affine transform, whereas STE's performance is erratic, often favoring simpler schemes.

| Configuration | Our Method Accuracy | | | STE Accuracy | | |
| | Linear | Affine x Lin W | Affine | Linear | Affine x Lin W | Affine |
|---|---|---|---|---|---|---|
| A1W1, Channel | 0.3335 | 0.3429 | **0.3442** | **0.3235** | 0.3085 | 0.3045 |
| A1W1, SCQ 128 | 0.3547 | 0.3706 | **0.3751** | 0.3399 | **0.3403** | 0.3397 |
| A4W1, Channel | **0.4016** | 0.4007 | 0.4000 | **0.3986** | 0.3971 | 0.3908 |
| A4W1, SCQ 128 | 0.4056 | 0.4066 | **0.4068** | 0.4042 | 0.4042 | 0.4036 |
| A4W4, Channel | **0.4219** | 0.4216 | 0.4218 | **0.4167** | 0.4142 | 0.4149 |
| A4W4, SCQ 128 | 0.4265 | 0.4258 | **0.4274** | 0.4245 | 0.4242 | **0.4254** |

Figure 6: Side-by-side comparison of quantization schemes (Linear, Affine X Linear W, and full Affine). **(Left)** With our denoising reconstruction method, a full Affine scheme provides the best performance especially at low precision. **(Right)** With the STE baseline, the benefit of an Affine scheme is inconsistent and sometimes detrimental, particularly at low precision.

### A.1.3 CENTERED HADAMARD TRANSFORM FOR OUTLIER MITIGATION

One effective strategy for mitigating outliers during quantization is to apply a change of coordinates that rotates the feature space, a technique designed to prevent large values in a single dimension from dominating the quantization grid. This approach, which we can call outlier blending, distributes the impact of an outlier across multiple new dimensions. This stands in contrast to the outlier localization philosophy of our main framework, which, as discussed in Section 6, uses subchannel quantization (SCQ) to confine an outlier's influence to a small, local block.

While early approaches considered random rotations (Tseng et al., 2024; Ashkboos et al., 2024; Liu et al., 2024), the randomness can be removed through the deterministic and computationally efficient Fast Hadamard Transform (Panferov et al., 2025). However, the deterministic structure of the Hadamard transform introduces a new challenge. Its first basis vector is a constant vector of all ones, creating a significant DC component. Real-world data in neural networks is often uncentered; when such data is transformed, this DC component can create a new, massive outlier in the rotated space, disrupting the subsequent quantization process.

To address this, we propose a novel application of our mean-correction technique to the Fast Hadamard Transform. We pre-center the data before applying the transform and add the mean back as a low-rank correction term afterward. The computation becomes:

$$(\boldsymbol{X} - \overline{\boldsymbol{x}} \cdot \boldsymbol{1}^T)\boldsymbol{H}\boldsymbol{H}^T(\boldsymbol{W} - \boldsymbol{1} \cdot \overline{\boldsymbol{w}}^T) + \overline{\boldsymbol{x}} \cdot \overline{\boldsymbol{w}}^T n \tag{8}$$

In our framework, we treat the Hadamard transform and Sub-Channel Quantization (SCQ) as *standalone* techniques to be compared against each other for improving quantization quality. For achieving ultimate quality, these techniques can be used together, but at a higher computational cost. Our analysis of the Pareto frontier curve revealed that SCQ with a block size of 128 consistently outperforms the quality gain from a standalone Hadamard transform. Furthermore, the Hadamard transform leads to extra energy costs not fully captured by our primary efficiency metric. Given these findings, we concluded that SCQ offers a more direct and efficient path to state-of-the-art performance. Therefore, while we validate our centered Hadamard approach here, we feature it in our storage efficiency study and prioritize SCQ for the main energy efficiency analysis.

This analysis highlights two distinct philosophies for handling the outlier dilemma (noise vs. feature). The rotation-based approach blends outliers into other coordinates, distributing their magnitude. Our framework, in contrast, blends outliers into the statistical parameters (scale and offset) of the dequantization transform by having all samples participate in the ridge regression (Eq. 3). This is made robust by SCQ, which ensures the statistical blending is a local effect. While our centered transform improves the Hadamard implementation (Table 3, Figure 7) , our main experiments

(Figure 2 show that the localized statistical blending via moderate-block-size SCQ consistently outperforms the gains from the global coordinate-blending of the Hadamard transform, confirming it as a more direct and effective solution.

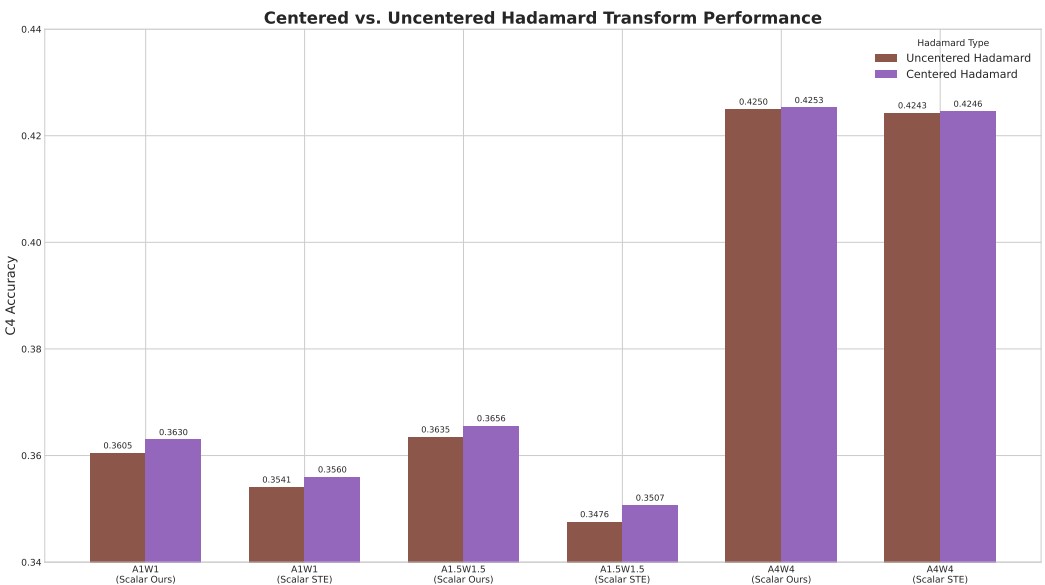

Figure 7: Comparison of applying the Hadamard transform to centered versus uncentered data. Pre-centering the data before the transform consistently improves accuracy across different bit-widths and for both our method and the STE baseline, confirming the benefit of mitigating the DC-induced outlier.

Table 3: Comparison of applying the Hadamard transform to centered versus uncentered data. Pre-centering the data consistently improves accuracy across different bit-widths for both our method and the STE baseline.

| Configuration | Method | Uncentered Acc. | Centered Acc. | Improvement |
|---|---|---|---|---|
| A1W1 | STE | 0.3541 | 0.3560 | +0.0019 |
| A1W1 | Our Method | 0.3605 | 0.3630 | +0.0025 |
| A1.5W1.5 | STE | 0.3476 | 0.3507 | +0.0031 |
| A1.5W1.5 | Our Method | 0.3635 | 0.3656 | +0.0021 |
| A4W4 | STE | 0.4243 | 0.4246 | +0.0003 |
| A4W4 | Our Method | 0.4250 | 0.4253 | +0.0003 |

### A.1.4 QUANTIZATION TO LOW-PRECISION FLOATS (FP4)

Our framework is not limited to integer quantization and also supports low-precision float formats. In this scenario, the quantization vector $q$ can represent values from grids like FP4 or FP8. Furthermore, our reconstruction parameters (scale and bias) are themselves resilient to low-precision storage; empirical evidence confirms that storing them in float8 (E5M2) does not adversely affect accuracy.

This flexibility allows for interesting comparisons between formats. We conducted an experiment to compare standard 4-bit integer quantization (INT4) against the E2M1 variant of 4-bit floating-point (FP4). We find that FP4 quantization yields a slightly better result than its INT4 counterpart. As shown in Table 4, the A4W4 model with our linear quantization achieves an accuracy of **0.4273** in FP4, compared to **0.4265** in INT4. We conjecture this is because FP4's format, which includes exponent bits, provides a wider dynamic range, making it inherently more tolerant to outliers than a uniform integer grid.

However, it is important to note that the best overall result in this configuration is still achieved by using an affine transform with integer quantization, which reaches an accuracy of **0.4274**. This

suggests that while FP4 is robust to outliers, explicitly modeling the data's asymmetry with an affine transform on a uniform grid can be an even more effective strategy.

Table 4: Comparison of 4-bit integer (INT4) and 4-bit floating-point (FP4) quantization using our method. FP4 slightly outperforms the standard linear INT4. However, the best performance is achieved by using an affine transform on the integer grid.

| Configuration | Quantization Scheme | Accuracy |
|---|---|---|
| A4W4, SCQ 128 | Linear (INT4) | 0.4265 |
| A4W4, SCQ 128 | Linear (FP4 E2M1) | 0.4273 |
| **A4W4, SCQ 128** | **Affine (INT4)** | **0.4274** |

### A.1.5 TRADE-OFFS IN STRUCTURED SPARSITY AND QUANTIZATION

Our framework enables a class of ultra-low precision models by combining M:N structured sparsity with quantization. This process creates ternary weights ($\{-1, 0, 1\}$) by first enforcing a sparsity mask and then quantizing the remaining non-zero values. For these approximately zero-centered ternary weights, we use the efficient linear reconstruction for the dequantization transform.

**Analysis of Efficiency Metrics**   When evaluating sparse models, it is crucial to consider not just storage but also computational energy.

- **Storage (Effective BPE):** The storage cost for M:N sparse formats must include metadata. For a block of 4 elements and 1-bit values, 1:4 sparsity requires 2 bits for metadata (a 2-bit index for the single non-zero location), yielding an effective BPE of $(1 + 2)/4 = 0.75$. In contrast, 2:4 sparsity requires a 4-bit mask (e.g., to encode a specific 2/4 pattern), resulting in a higher BPE of $(2 + 4)/4 = 1.5$.

- **Computation (Energy Score):** The true benefit of structured sparsity lies in computational savings. Modern hardware can exploit N:M patterns to reduce the number of multiply-accumulate operations. We estimate this with an energy score (Sparsity Factor × Act. Bits × Weight Bits). For 2:4 sparsity, this factor is $2/4 = 0.5$, halving the computational cost.

**Results and Analysis**   As shown in Table 5, combining structured sparsity with our method reveals important trade-offs between accuracy, storage, and energy.

Our sub-1-bit model (1:4 sparsity) provides a compelling option for memory-constrained environments. It achieves a $25\%$ reduction in weight storage (0.75 BPE) and a $75\%$ reduction in computational energy, with only a minor and graceful degradation in accuracy compared to the dense A4W1 baseline.

Most notably, the 2:4 sparse model achieves the highest accuracy of all (0.4080), slightly outperforming the dense baseline. While its storage footprint is higher (1.5 BPE), its primary advantage is a $50\%$ reduction in energy and compute cost. This demonstrates that 2:4 structured sparsity introduces a powerful representation that can simultaneously boost model accuracy and dramatically lower inference energy consumption.

Table 5: Comparison of dense and sparse weights on Gemma 1B (A4W1). Structured sparsity provides significant energy savings. The 2:4 model halves the energy cost while slightly improving accuracy.

| Weight Type | Sparsity | Accuracy | Effective BPE | Energy Score |
|---|---|---|---|---|
| **Dense Baseline** | **None** | **0.4056** | **1.00** | **4.0** |
| Sparse Ternary | 1:4 | 0.3979 | 0.75 | 1.0 |
| Sparse Ternary | 2:4 | 0.4080 | 1.50 | 2.0 |

### A.1.6 THE ENERGY EFFICIENCY FRONTIER: METHODOLOGY

To meaningfully evaluate the benefits of our quantization and sparsification framework, we must move beyond conventional metrics like FLOPs and instead estimate the total computational cost. We adopt a hardware-inspired, energy-aware metric motivated by the principle that on modern ML hardware, the energy consumed by a multiply-accumulate (MAC) operation is roughly proportional to the product of the bit-widths of its operands (Zhang et al., 2022). This is because the fundamental cost of a digital multiplier scales quadratically with the precision of the numbers it processes.

Following this principle, we define an **Approximate Total Energy Cost** score for each model configuration. This score serves as a hardware-agnostic proxy for the total energy consumed by the arithmetic operations that dominate the computational workload.

The score is calculated as follows:

$$\text{Total Energy Cost} \approx (\text{Sparsity Factor}) \times (\text{Activation Bits}) \times (\text{Weight Bits}) \times (\text{Total Operations}) \tag{9}$$

Where:

- **Activation Bits** and **Weight Bits** are the precisions used for the activations and weights, respectively. The product, Activation Bits $\times$ Weight Bits, represents the relative energy cost of a single MAC operation. For a standard BF16 operation, this baseline cost is $16 \times 16 = 26$, whereas an A4W1 operation has a relative cost of $4 \times 1 = 4$—a 64x reduction in approximate energy per operation.

- **Sparsity Factor** accounts for the reduction in MAC operations due to structured sparsity. For M:N sparsity (e.g., 2:4), this factor is $\frac{M}{N}$. For a 2:4 sparse model, the computational cost is effectively halved, so the sparsity factor is 0.5. For a dense model, this factor is 1.0.

- **Total Operations** is the total number of multiply-accumulate (MAC) operations in the model. This factor scales the per-operation cost to reflect the model's overall size and computational density. When comparing models of different scales, we use the total number of parameters as a proxy for the total operations.

It is worth noting that some techniques, such as the Hadamard transform, introduce additional computational costs not captured by this MAC-centric metric. While the Hadamard transform has a complexity of $O(n \log n)$, for simplicity, we omit a detailed analysis of its energy contribution in this study. This metric nevertheless provides a unified and principled foundation to systematically map the trade-offs between total energy efficiency and model accuracy. By using this score, we can construct the energy efficiency Pareto frontier shown in Figures 3, 4, revealing the synergistic power of combining our robust training framework with asymmetric quantization and structured sparsity.

Table 6: Comparison of Gemma 1B vs. 4B models. A larger model (4B) aggressively quantized with our method surpasses the performance of a smaller (1B) full-precision model.

| Model | Act Bits (A) | Weight Bits (W) | SCQ | Sparsity | Method | Accuracy |
|-------|------------|-----------------|-----|----------|--------|----------|
| Gemma3 1B | 16 | 16 | - | None | Baseline | 0.4494 |
| Gemma3 1B | 4 | 4 | 128 | None | Affine | 0.4443 |
| Gemma3 4B | 16 | 16 | - | None | Baseline | 0.4706 |
| Gemma3 4B | 1 | 1 | 128 | None | Affine | 0.4221 |
| Gemma3 4B | 1.5 | 1.5 | 128 | None | Linear | 0.4298 |
| Gemma3 4B | 1.5 | 1.5 | 32 | None | Linear | 0.4393 |
| Gemma3 4B | 2 | 2 | 128 | None | Affine | 0.4465 |
| Gemma3 4B | 4 | 1 | 128 | None | Linear | 0.4496 |
| Gemma3 4B | 4 | 1 | 128 | None | Affine | 0.4510 |
| Gemma3 4B | 4 | 1 | 128 | 2:4 | Linear | 0.4517 |

Table 7: Top-1 Validation Accuracy of ResNet-50 on the ImageNet Dataset.

| Precision | 100 Epochs | 400 Epochs |
|-----------|-----------|-----------|
| A32W32 | 76.41 | – |
| A4W4 | **76.45** | – |
| A4W2 | 75.12 | 75.59 |
| A4W1 | 72.04 | 73.97 |

### A.1.7 RESNET-50 ON IMAGENET

We utilized the Flax framework to train ResNet-50 from scratch on ImageNet, employing stochastic gradient descent with an initial learning rate of 0.1, training the model for 100 epochs with weight decay of 0.0001. As shown in Table 7, the top-1 accuracy from the A4W4 configuration (76.45) surpasses the baseline (76.41) without any hyperparameter tuning. This demonstrates the effectiveness of our method in achieving competitive performance without requiring extensive optimization. We compared our results to previously reported A4W4 quantization results for ResNet-50 trained on ImageNet. Our results compare favorably to existing work, without the need for additional operations such as parameter search, fine-tuning, calibration, clipping, gradient estimation, or reinforcement learning (Table 8).

Table 8: Comparison of top-1 accuracy for A4W4 ResNet-50. The columns represent: FP32: Full precision baseline. A4W4: Quantizing both activations and weights to 4 bits. GE: Whether gradient estimation is involved. PT: Pretraining/finetuning/calibration required. Clip: Clipping required. LB: The lowest bitwidth reported in the corresponding paper. [*]Estimated from Fig. 1 (Abdolrashidi et al., 2021)

| Method | FP32 | A4W4 | GE | PT | Clip | LB |
|--------|------|------|----|----|------|----|
| AQT (Abdolrashidi et al., 2021) | 76.65 | 76.4[*] | Y | Y | Y | 4 |
| VS-Quant (Dai et al., 2021) | 76.16 | 75.28 | Y | Y | Y | 3 |
| FAQ (McKinstry et al., 2018) | 76.15 | 76.25 | Y | Y | Y | 4 |
| HAQ (Wang et al., 2019) | 76.15 | 76.14 | N | Y | Y | 4 |
| **Ours** | **76.41** | **76.45** | **N** | **N** | **N** | **1** |

### A.1.8 TRANSFORMER ON WMT

To evaluate the effectiveness of our method on transformer models, we employed the Flax framework to train the transformer model on two WMT2017 datasets (EN-DE, DE-EN) and subsequently assessed its performance on the corresponding WMT2014 datasets. The training process utilized the AdamW optimizer with weight decay set to 0.1 and a cosine scheduler for 25,000 steps, employing a batch size of 1024. Recognizing the known slow convergence of transformer models, we extended the training duration to 100,000 steps (Table 9). Remarkably, our low-precision results consistently surpass the full-precision baseline.

Given the prevalence of transformers in large language models, extensive research has been dedicated to quantizing transformer models. We compare our findings to other works, and ours stands out as the only method that can surpass the full-precision baseline (Table 10). This achievement highlights the unique strength of our formulation, which not only preserves signal fidelity but also benefits from regularization effects. Several recent works have explored alternative quantization approaches using different datasets, which are not included in this table. One such method is AWQ, a weight-only quantization 4-bit quantization method (Lin et al., 2023), requires retaining $1\%$ of salient weights and all activations unquantized. Their method also involves searching for an optimal scaling factor and a calibration set. Additionally, BitNet (Wang et al., 2023), presents a 1-bit quantization method for transformers. The lowest activation precision achieved in their work is 8 bits, exceeding the highest activation bit in our method. Their method also necessitates clipping, additional normalization, and recipe changes.

Binary Transformers

Table 9: BLEU Score of training low-precision Transformers on the WMT datasets. Models trained for 100k steps consistently outperform their 25k counterparts, and low-precision settings like A4W4 and A4W2 achieve results competitive with or even exceeding the full-precision baseline.

| Precision | DE-EN | | EN-DE | |
|---|---|---|---|---|
| | 25k Steps | 100k Steps | 25k Steps | 100k Steps |
| A32W32 | 33.5 | 33.9 | 29.49 | 29.8 |
| A4W4 | 33.78 | 33.64 | 29.71 | **30.17** |
| A4W2 | 33.45 | **34.04** | 28.58 | 30.03 |
| A4W1 | 32.76 | 33.66 | 27.06 | 28.32 |
| A2W2 | 32.32 | 33.51 | 27.56 | 28.61 |
| A2W1 | 31.39 | 32.51 | 26 | 27.4 |
| A1W1 | 27.4 | 28.27 | 21.42 | 23.64 |

Table 10: BLEU score comparison of A4W4 transformers. Columns are defined as in Table 8

| Method | FP32 | A4W4 | GE | PT | Clip | LB |
|---|---|---|---|---|---|---|
| LSQ+LUQ (Xi et al., 2023) | 27.5 | 27.17 | Y | N | Y | 4 |
| Fixed-Point (Boo & Sung, 2020) | 28.48 | 26.94 | Y | Y | Y | 4 |
| GradScale (Sun et al., 2020) | 27.5 | 25.9 | Y | N | N | 4 |
| LUQ+SMP (Chmiel et al., 2021) | 27.5 | 27.25 | N | Y | Y | 4 |
| **Ours** | **29.49** | **29.71** | **N** | **N** | **N** | **1** |

Table 11: BLEU Score of Transformers with binary activations on the WMT datasets. For the last column we record the drop from full precision models.

| Precision | DE-EN | | EN-DE | |
|---|---|---|---|---|
| | 25k Steps | 100k Steps | 25k Steps | 100k Steps (Drop %) |
| A1W4 | 29.74 | 30.74 | 24.07 | 26.28 (-11.81%) |
| A1W2 | 28.81 | 29.81 | 23.4 | 25 (-16.11%) |
| A1W1 | 27.4 | 28.27 | 21.42 | 23.64 (-20.67%) |
| A1W1 (Zhang et al., 2023) | – | – | – | 17.87 (-32.18%) |

Inspired by the temporal nature of the transformer model, we reduced the activation precision to 1-bit for all linear layers, transforming it into a temporal binary network, akin to a quasi-spiking neural network. However, unlike traditional spiking neural networks, our model doesn't simulate spike generation or consider spiking frequency. To evaluate our approach, we assigned 1, 2, and 4 bits to the weights (Table 11) . Our results showcase that these converted binary transformers remain highly competitive with full-precision counterparts . A recent study attempted to binarize transformers (Zhang et al., 2023). Their approach included extra normalization layers, clipping, and progressive quantization during training . We compared our method to their A1W1 configuration, achieving significant improvements (Table 11, last column) .

A.1.9 SUBCHANNEL BLOCK SIZE

In addition to precision and sparsity, subchannel block size also presents a trade-off between accuracy and efficiency. We observe that its influence is more pronounced at extremely low precision levels, such as 1-bit. While using a smaller block size can improve performance, the effective bits per element can easily exceed the original design. We provide some comparisons here. Considering this trade-off, lower precision models may not always be more efficient (Table 12). Firstly, the accuracy achieved with smaller blocks in an A1W1 models may not surpass the accuracy achieved using A2W2. The perturbations introduced during lower precision training often remain high, hindering the achievement of high quality. Optimal selection needs to be made based on the underlying

hardware support and the problem of interest. Our work facilitates a comprehensive study of this trade-off.

Table 12: BLEU Score comparison of adjusting the block size when training the A1W1 Transformers for 100k steps on the WMT DE-EN Dataset.

| Block Size | A1W1 BLEU (100k) |
|:---:|:---:|
| 32 | 29.71 |
| 128 | 28.27 |
| 512 | 27.14 |

### A.1.10 THE DUAL ROLE OF THE DENOISING TRANSFORM AND REGULARIZATION ($\lambda$)

The stability of our framework arises from the synergy between two components: the novel gradient path created by our dequantization transform and the regularization term $\lambda$. This section aims to disentangle these two effects and clarify their respective roles.

The primary innovation of our method is the new gradient path, $\frac{dg(\boldsymbol{q})}{d\boldsymbol{q}}$, which makes the backward pass "quantization-aware." This path is the fundamental mechanism that allows the network to learn to be robust to quantization error by forcing the error term $\delta$ to participate in the gradient computation. This mechanism exists even when $\lambda = 0$. However, this new gradient path can be numerically unstable. The denominator of the scaling factor $s_g$ in our transform (Eq. 3 and its linear counterpart) contains the term $Var_q$. In common scenarios where a block of data has zero or near-zero variance—for example, if all activations after a ReLU are positive and get quantized to the same integer—this denominator collapses, leading to division by zero and training divergence. This is precisely the behavior observed in Table 13 when $\lambda = 0$.

The regularization term $\lambda$ provides the principled solution to this instability. It is not an ad-hoc fix but an integral part of the ridge regression objective (Eq. 2 and 4) from which our transform is derived. Ridge regression is specifically designed to provide stable solutions to such ill-conditioned problems. The $\lambda$ term ensures the denominator is always bounded away from zero, guaranteeing a stable and well-defined solution for the scaling factor $s_g$. While a simple heuristic like clipping the variance (e.g., $max(Var_q, \epsilon)$) could also prevent NaNs, our use of $\lambda$ is more theoretically sound as it smoothly discounts the influence of noisy, low-variance blocks.

As described in Section 3.3, $\lambda$ also acts as a "denoising" knob. A larger $\lambda$ suppresses the impact of perturbations by reducing the scale $s_g$, forcing the transform to rely more on the stable mean of the original signal, $\overline{\boldsymbol{x}}$.

In our experiments, we observe that a wide range of $\lambda$ values yield satisfactory results (Table 13). For higher precision ($\geq 4 - bits$), where quantization noise is lower, smaller $\lambda$ values (e.g., 0.0001) can be safely used. However, to ensure stability across all settings, especially during the transition to 1-bit quantization, we set our preference on the safer side and use a default of $\lambda = 0.01$.

In summary, the stability of our framework comes from a two-part design: the new gradient path provides the mechanism for error-aware learning, while the $\lambda$ regularization provides the principled robustness that makes this mechanism numerically stable and viable in practice.

Table 13: BLEU Score comparison of adjusting the $\lambda$ when training the A1W1 Transformers for 25k steps on the WMT EN-DE Dataset.

| $\lambda$ | A1W1 BLEU (25k) |
|:---|:---:|
| 1.0 | 5.83 |
| 0.01 | 21.42 |
| 0.0001 | 20.08 |
| 0 | NaN |

A.1.11 ADDITIONAL EXPERIMENTAL VALIDATION

In this section, we provide further empirical evidence to substantiate the robustness of our method across different scales and distinct quantization schemes.

**Scalability to OpenWebText (GPT-2 Small).** To validate that our stability findings generalize to standard language modeling benchmarks, we trained a GPT-2 Small model on the OpenWebText dataset. Figure 8 illustrates the training loss trajectories in the challenging A1W1 regime. Our method demonstrates smooth and stable convergence throughout the training process. In contrast, the standard STE baseline and BitNet exhibit significant instability or failure to converge to a competitive loss. This confirms that the robustness of our denoising estimator holds as model capacity and dataset complexity increase.

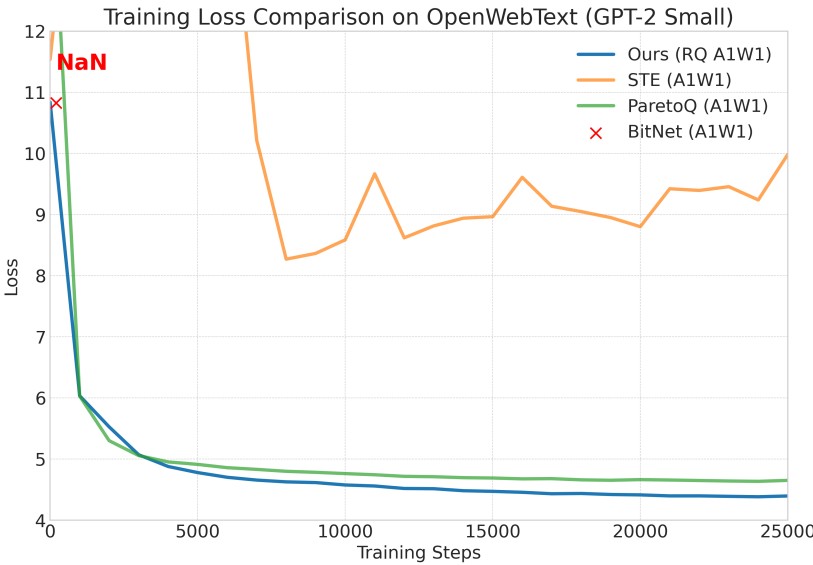

Figure 8: **Scalability: GPT-2 Small on OpenWebText (A1W1).** Our method (Blue) converges smoothly, whereas STE (Orange) and BitNet (Red Cross) exhibit instability or divergence.

**Isolating Estimator Gains: Linear vs. Linear Comparison.** To rigorously isolate the contribution of our denoising estimator from the benefits of affine quantization (addressing Reviewer BVSw's inquiry), we reproduced the Shakespeare A1W1 experiment with additional "linear-only" configurations. We compared our method against the standard STE, BitNet, and ParetoQ, restricting all methods to identical symmetric linear quantization schemes where applicable.

The results, presented in Figure 9, demonstrate three critical findings :

1. **Estimator Stability (Linear vs. Linear):** When restricted to linear quantization, our method converges stably to a low validation loss ($\approx$ 2.1). In stark contrast, the standard STE with linear quantization exhibits significant instability and diverges to a high loss ($>$ 5.0). This confirms that our estimator provides a fundamental stability advantage independent of the quantization mapping.

2. **Comparison with Strong Baselines:** Our linear implementation outperforms both BitNet and ParetoQ in terms of convergence speed and final validation loss.

3. **The Affine Advantage:** While our linear method is robust, enabling our full affine quantization yields the lowest overall loss ($\approx$ 1.9). This reinforces that our estimator is robust enough to unlock the additional expressivity of affine parameters, which other methods fail to utilize effectively.

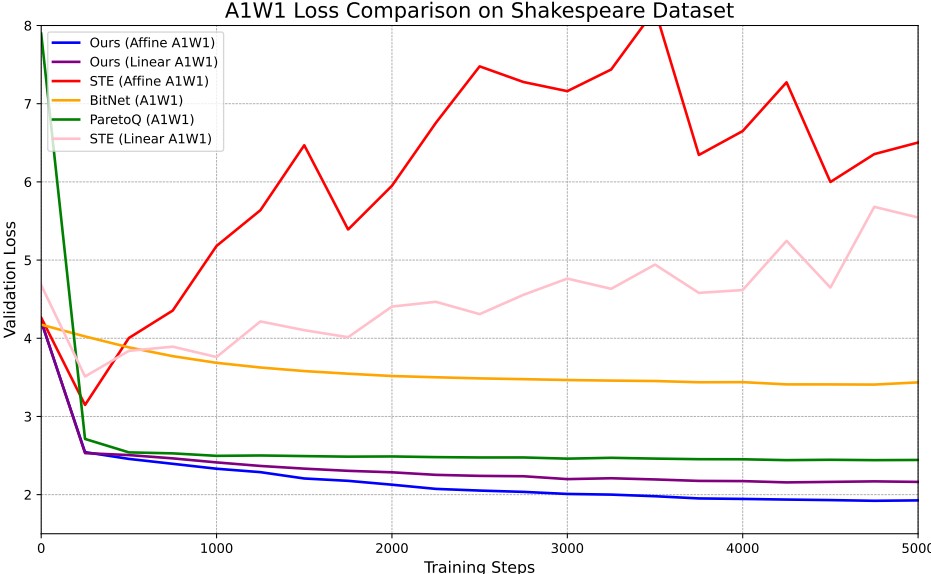

Figure 9: **Mechanism Isolation: Linear-vs-Linear Ablation (Shakespeare).** We isolate the estimator's effect by running "Ours" and "STE" with identical linear schemes. **Ours (Linear)** remains stable and outperforms **STE (Linear)**, confirming the estimator's intrinsic value. **Ours (Affine)** achieves the best overall performance.

## A.2 Computational Flow for Quantized Matrix Multiplication

The practical application of our method during training and inference follows one of two computational paths, depending on the availability of specialized hardware for low-precision arithmetic.

**Simulation on Standard Hardware (Fake Quantization).** In the absence of native hardware support for low-precision integer matrix multiplication (e.g., on standard CPUs or GPUs), the benefits of quantization are simulated. This "fake quantization" flow is beneficial for Quantization-Aware Training (QAT) and for validating model accuracy. The process involves quantizing the high-precision input tensors (weights and activations) and then immediately dequantizing them back to a floating-point format. This injects the quantization error into the tensors, which are then multiplied using standard floating-point arithmetic. While this approach correctly models the accuracy impact of quantization, it does not yield the computational or energy savings of true low-precision matrix multiplication.

**Native Low-Precision Inference with Hardware Support.** When specialized hardware is available (e.g., TPUs or modern GPUs with integer matrix-multiply units), the full efficiency of our framework can be realized. In this flow, the input tensors are quantized to low-precision integers and the core matrix multiplication, $Q_X \cdot Q_W$, is performed directly using highly efficient integer arithmetic. The result, still in a low-precision integer format, is then dequantized back to the floating-point domain using our shortcut formula. This final dequantization step, which includes the low-rank corrections and scaling, is performed in floating-point but is computationally inexpensive compared to the main matrix multiplication. This two-stage process—a fast, low-precision multiply followed by a cheap, high-precision dequantization—unlocks the full performance and energy benefits of our method.

## A.3 Complexity Analysis of the Affine Shortcut

The primary benefit of the novel affine shortcut (Theorem 1, Section 5) is the reduction of computational overhead from the naive expansion of the two-sided affine quantized matrix multiplication. Here, we analyze the complexity of the shortcut formula:

$$\tilde{Y} = (\boldsymbol{s}_X \cdot \boldsymbol{s}_W^T) \odot (Q^X \cdot Q^W - \overline{\boldsymbol{q}}_X \cdot \overline{\boldsymbol{q}}_W^T n) + \overline{\boldsymbol{x}} \cdot \overline{\boldsymbol{w}}^T n$$

Let the dimensions of the input matrix $X$ be $M \times N$ and the weight matrix $W$ be $N \times P$. The output matrix $Y$ is $M \times P$. The variables $s_X, \overline{q}_X, \overline{x}$ are row-wise statistics ($M \times 1$), and $s_W, \overline{q}_W, \overline{w}$ are column-wise statistics ($P \times 1$).

1. **Main Computation:** The core term is the low-precision integer matrix multiplication (Int-MM): $Q^X \cdot Q^W$.
   - **Complexity:** $\mathcal{O}(MNP)$. This is the cost of the standard matrix multiplication, performed in high-speed, low-precision arithmetic.

2. **Correction Terms and Element-Wise Operations:** The remaining operations constitute the overhead:
   - **Statistical Overhead:** Calculating the mean ($\overline{q}_X, \overline{q}_W, \overline{x}, \overline{w}$) and scale ($s_X, s_W$) terms requires computing the statistics of $X$ and $W$. This involves $\mathcal{O}(MN + NP)$ complexity, a negligible cost compared to the main matrix multiplication.
   - **Rank-1 Corrections:** The two terms $\overline{q}_X \cdot \overline{q}_W^T n$ and $\overline{x} \cdot \overline{w}^T n$ are rank-1 outer products, which are then scaled. Their complexity is $\mathcal{O}(MP)$.
   - **Element-Wise Scaling:** The final element-wise (Hadamard) multiplication $\odot$ and the additions/subtractions all have complexity $\mathcal{O}(MP)$.

**Conclusion on Overhead** The total overhead introduced by the shortcut is dominated by the $\mathcal{O}(MP)$ complexity of the low-rank corrections and the $\mathcal{O}(MN + NP)$ complexity of the statistical calculations. Since the main matrix multiplication has a complexity of $\mathcal{O}(MNP)$, **the relative overhead of the affine shortcut is negligible** in the typical setting where the depth $N$ is large (i.e., $N \gg M$ and $N \gg P$).

The shortcut's complexity is asymptotically equivalent to a standard low-precision linear quantized matrix multiplication, making the robust, high-quality affine quantization practically efficient.

### A.4 GRADIENT ANALYSIS: EXPLICIT DEPENDENCE ON QUANTIZATION ERROR

In Section 3.3.2, we posit that our denoising transform establishes a backward pass that is explicitly aware of the quantization error $\delta$, addressing the "blind spot" of the standard Straight-Through Estimator (STE). Here, we provide a rigorous derivation of the local gradient (Jacobian) to validate this claim.

Consider the symmetric (linear) dequantization case for simplicity (Eq. 4). The transform is defined as:
$$g(\boldsymbol{q}) = s_g(\boldsymbol{q}) \cdot \boldsymbol{q} \tag{10}$$
where the scale factor $s_g$ is not a constant parameter, but a scalar function of the quantized vector $\boldsymbol{q}$ derived from the ridge regression solution: $s_g(\boldsymbol{q}) = \frac{\boldsymbol{x}^\top \boldsymbol{q}}{\boldsymbol{q}^\top \boldsymbol{q} + \lambda}$.

To analyze the backward pass, we compute the Jacobian matrix $\boldsymbol{J} = \frac{\partial g(\boldsymbol{q})}{\partial \boldsymbol{q}}$. Applying the product rule to $g(\boldsymbol{q})$ yields:
$$\frac{\partial g(\boldsymbol{q})}{\partial \boldsymbol{q}} = s_g(\boldsymbol{q}) \cdot \mathbf{I} + \boldsymbol{q} \cdot (\nabla_{\boldsymbol{q}} s_g(\boldsymbol{q}))^\top \tag{11}$$
where $\mathbf{I}$ is the identity matrix.

The first term, $s_g(\boldsymbol{q}) \cdot \mathbf{I}$, corresponds to a standard scaling operation. The second term, however, introduces a data-dependent rank-1 correction. We derive $\nabla_{\boldsymbol{q}} s_g(\boldsymbol{q})$ using the quotient rule:
$$\nabla_{\boldsymbol{q}} s_g = \frac{(\boldsymbol{q}^\top \boldsymbol{q} + \lambda)\boldsymbol{x} - (\boldsymbol{x}^\top \boldsymbol{q})2\boldsymbol{q}}{(\boldsymbol{q}^\top \boldsymbol{q} + \lambda)^2} \tag{12}$$

Substituting this result back into the Jacobian expression reveals that the gradient is a complex, non-linear function of $\boldsymbol{q}$.

**The Role of Quantization Error.** Since $\boldsymbol{q} = f(\boldsymbol{x}) + \boldsymbol{\delta}$, the local derivative $\frac{\partial g}{\partial \boldsymbol{q}}$ is explicitly parameterized by the quantization error $\boldsymbol{\delta}$.

**Comparison with STE.** In standard STE, the backward pass typically assumes an identity (or constant scalar) derivative $\frac{\partial g(q)}{\partial x} \approx \mathbf{I}$, effectively rendering the gradient independent of the specific value of $\boldsymbol{\delta}$. In contrast, our method computes the gradient of the loss with respect to $\boldsymbol{x}$ via the chain rule:

$$\frac{\partial \mathcal{L}}{\partial \boldsymbol{x}} = \frac{\partial \mathcal{L}}{\partial g(\boldsymbol{q})} \cdot \underbrace{\frac{\partial g(\boldsymbol{q})}{\partial \boldsymbol{q}}}_{\text{Error-Aware}} \cdot \frac{\partial \boldsymbol{q}}{\partial \boldsymbol{x}} \tag{13}$$

Because the term $\frac{\partial g(q)}{\partial q}$ varies based on the magnitude and direction of the noise $\boldsymbol{\delta}$ (embedded within $\boldsymbol{q}$), the learning signal flowing back to the weights is dynamically modulated by the instantaneous quantization error. This mechanism is mathematically analogous to the stabilizing gradients found in Normalization layers, forcing the network to adapt to the noisy quantization surface.

## A.5 RELATED WORK

Neural network quantization has become a widely adopted technique for reducing memory footprint and accelerating inference time, enabling efficient deployment on resource-constrained devices (Gholami et al., 2022). While full-precision models typically store weights in floating-point format, quantized weights are represented as integers, typically using 8 bits (Dai et al., 2021; Wortsman et al., 2023; Jacob et al., 2018), 3-4 bits (Dettmers et al., 2023; Liu et al., 2021b; Abdolrashidi et al., 2021; Dai et al., 2021), or even 1 bit (Zhang et al., 2022; Liu et al., 2021a; Wang et al., 2023; Zhang et al., 2023; Courbariaux et al., 2016; Rastegari et al., 2016). In addition to quantizing weights, model activations can also be quantized to further enhance computational efficiency (Dai et al., 2021; Jacob et al., 2018; Esser et al., 2019).

Although 8-bit quantization is commonly used as a standard practice in industry (Jacob et al., 2018), achieving lower-bit quantization remains challenging and requires specialized techniques to ensure robust training. Several common techniques include: 1. Mixed precision quantization: This approach selectively assigns different bit levels to different weights, aiming to optimize the trade-off between model size and accuracy (Wang et al., 2019; Lin et al., 2023; Han et al., 2015; Défossez et al., 2021). 2. Training recipes: These techniques compensate for the discontinuities introduced by quantization by employing strategies such as sharpness-aware minimization (Liu et al., 2021b; Foret et al., 2020), state-aware optimization (Liu et al., 2021a), knowledge distillation (Kim et al., 2019), and multi-phase training (Liu et al., 2021c). 3. Quantization-friendly network architectures: This approach involves replacing original network layers with alternatives that are more amenable to quantization (Zhang et al., 2022).For example, Bi-Real Net enhances 1-bit CNNs by adding an identity shortcut to preserve some real-valued information in the forward pass Liu et al. (2018).

In contrast to prior work, our method explicitly models quantization discontinuities as perturbations. We decompose the perturbed signal into clean and noisy components, then apply denoising to suppress the noise. This approach leads to a closed-form solution that guarantees training convergence even at extremely low bitwidths. While previous methods have also modeled quantization noise using continuous distributions (e.g., Uniform or Gaussian) for gradient estimation (Défossez et al., 2021; Ballé et al., 2016), they do not optimize the reconstruction process itself to enhance training stability.

To further reduce model footprint, researchers have been combining sparsity/pruning and quantization in a unified formulation to further compress neural networks (Park et al., 2022; Yang et al., 2020). In this paper, we extend our noise injection and denoising reconstruction theory to sparsity.

In the time since our method was first proposed, the community has made rapid advancements in low-bit LLM training. One major thrust has been to improve the Quantization-Aware Training (QAT) process itself. Recent works like ParetoQ (Liu et al., 2025) have developed unified frameworks to systematically compare different bit-widths and highlight the importance of the quantization function design. By optimizing training schemes and refining quantization functions, ParetoQ surpasses methods tailored to specific bit-widths and reveals that ternary and 2-bit quantization can exceed 4-bit performance in the size-accuracy trade-off.

Concurrently, methods like QuEST (Panferov et al., 2025) have sought to improve upon the standard STE by designing more sophisticated backward passes. QuEST introduces a "trust gradient estimator" to stabilize training, coupled with a Hadamard normalization in the forward pass to make weight

distributions more suitable for quantization. Another direction has focused on combining binarization with structured sparsity to break the 1-bit barrier. Post-training methods like STBLLM achieve effective bit-widths as low as 0.55 by employing N:M sparsity techniques Dong et al. (2024).

Our work addresses the core problem of training instability from a different, foundational perspective. Instead of developing bit-specific heuristics or entirely new gradient estimators, we explicitly model the quantization discontinuity as an additive perturbation. Our primary contribution is a denoising dequantization transform, derived from a principled objective, that makes the training process inherently aware of and robust to this quantization error. By deriving well-defined gradients without surrogates, our approach offers a theoretically grounded alternative to the heuristic, gradient estimation-based methods.

## A.6 REFERENCE CODE

We provide the reference code for our core contributions.

```python
def affine_quantize(x, bits, axis, eps=1e-8):
    """Quantizes a tensor to the range [0, 2**bits - 1]."""
    max_val = jnp.max(x, axis=axis, keepdims=True)
    min_val = jnp.min(x, axis=axis, keepdims=True)

    # Scale to the target integer range
    scaled_x = (x - min_val) / (max_val - min_val + eps) * (2**bits - 1)

    # Detach rounding error from the gradient
    delta = jax.lax.stop_gradient(jnp.round(scaled_x) - scaled_x)

    # Inject quantization error
    q = scaled_x + delta
    return q
```

Code Snippet 1: JAX reference code for quantization: $q = f(x) + \delta$.

```python
def affine_dequant(q, x, axis, lmd=1e-2):
    """Dequantizes based on a ridge regression objective."""
    E_q2 = jnp.mean(q**2, axis=axis, keepdims=True)
    E_q  = jnp.mean(q,    axis=axis, keepdims=True)
    E_qx = jnp.mean(q * x,axis=axis, keepdims=True)
    E_x  = jnp.mean(x,    axis=axis, keepdims=True)

    Var_q = E_q2 - E_q**2
    Cov_qx = E_qx - E_q * E_x

    # Solve for scale (s) and offset (o)
    s = Cov_qx / (Var_q + lmd)
    # o = E_x - s * E_q

    return s * (q - E_q) + E_x # Reconstruct: r = s*q + o
```

Code Snippet 2: The denoising dequantization transform $r = g(q)$

## B ACKNOWLEDGEMENTS

We extend our sincere gratitude to Abhijit Ogale, Dinghua Li, Jeff Dean, Jian Li, Kyuyeuan Kim, Rasmus Larsen, Sameer Agarwal, Sanjiv Kumar, Sergey Ioffe, Tammo Splank, Zhifeng Chen for their insights and supports.

```python
def affine_quantized_matmul_shortcut(x, w, bits, lmd=1e-2):
    """Computes an efficient affine quantized matmul."""
    # Note: Assumes a function affine_quant that returns (q, s, o)
    q_x, s_x, _ = affine_quant(x, bits=bits, axis=-1)
    q_w, s_w, _ = affine_quant(w, bits=bits, axis=0)

    # Decompose into linear term + two rank-1 corrections
    linear_term = q_x @ q_w
    corr_term_1 = q_x.mean(-1, keepdims=True) @ q_w.sum(0, keepdims=True)
    corr_term_2 = x.mean(-1, keepdims=True) @ w.sum(0, keepdims=True)

    # Combine terms as per Theorem 1
    return (s_x * s_w) * (linear_term - corr_term_1) + corr_term_2
```

Code Snippet 3: The shortcut formula for affine quantized matrix multiplication

