# OpenReview forum: "Robust Training of Neural Networks at Arbitrary Precision and Sparsity"
_ICLR.cc/2026/Conference — ICLR 2026 Poster_

### Official Review · Reviewer_BVSw · 2025-10-29

**Soundness:** 3
**Presentation:** 3
**Contribution:** 3
**Rating:** 6
**Confidence:** 2

**Summary:**

The paper locates the root cause of QAT instability in the STE’s "quantization-oblivious" backward pass and replaces it with a non-STE framework. The core method is a denoising dequantization transform derived from a ridge-regression objective, which injects an explicit corrective gradient path for quantization error and extends naturally to sparsification. The authors also introduce a shortcut for efficient affine-quantized matmul. The framework supports training across a wide range of precisions, including fully binary (A1W1) and sparse sub-1-bit models, with standard recipes.

**Strengths:**

1. A ridge-regularized dequantization transform creates an error-aware gradient path, stabilizing training at ultra-low precision and treating sparsification as quantization within one framework.
2. The approach is compatible with standard autograd and includes a shortcut formula that makes affine-quantized matmul computationally viable in practice.
3. At low precisions, the method achieves a new storage–energy Pareto frontier (demonstrated on Gemma-1B), offering clear, actionable trade-off guidance for model design.

**Weaknesses:**

1. Efficiency claims rely on a hardware-agnostic energy proxy rather than end-to-end measurements (latency/throughput/energy) on real accelerators.

2. Using affine+SCQ for the proposed method versus symmetric implementations for BitNet/ParetoQ can over-attribute gains to the estimator; require matched schemes (linear vs. linear) to isolate estimator effects.

**Questions:**

1. Can you clarify whether, in the extreme A1W1 regime, memory bandwidth remains the dominant bottleneck or whether the additional “shortcut” terms shift execution to a compute-bound regime—and, if so, whether these computations can be overlapped (e.g., via pipelining or operator fusion) with the main integer GEMM/memory traffic to hide their latency?
2. Can you provide ablations using linear (symmetric) quantization only, where your method and all baselines (STE, BitNet, ParetoQ) use the same quantization scheme and SCQ block size, so the estimator’s contribution is isolated from affine/SCQ effects?
3. Can you report how your method scales beyond 1B and 4B (e.g., ≥7B/13B/30B) and what advantage—if any—it holds over BitNet and related A1W1 approaches?

---

> ### Author Response · Authors · 2025-11-19
>
> We thank Reviewer BVSw for their excellent, detailed questions and for appreciating our method's ability to unify sparsification and quantization and create a new Pareto frontier.
>
> **Weakness 1:** Efficiency claims rely on a hardware-agnostic energy proxy rather than end-to-end measurements...
>
> **Response:** This is a valid limitation. We used this proxy (Appendix A.1.6) as it is a "principled, hardware-agnostic" first-order approximation of arithmetic cost, which scales with $A_{bits} \times W_{bits}$. This is standard for comparing the relative efficiency of different bit-width/sparsity configurations, which was our goal. We will strengthen the acknowledgement in the text that this proxy omits data movement and other costs, and that end-to-end hardware measurement, while a significant engineering task beyond this paper, is the critical next step to validate the practical gains.
>
> **Weakness 2 & Question 2:** Using affine+SCQ for the proposed method versus symmetric implementations for BitNet/ParetoQ can over-attribute gains... require matched schemes (linear vs. linear)...
>
> **Response:** We thank the reviewer for this critical suggestion. We agree that a "linear-vs-linear" comparison is necessary to strictly isolate the benefit of our denoising estimator from the expressivity gains of affine quantization. During the rebuttal, we conducted a controlled reproduction of the A1W1 experiment on the Shakespeare dataset, restricting our method and all baselines (STE, BitNet, ParetoQ) to identical symmetric linear quantization schemes.
>
> **New Results (Please see Additional Experimental Validation Fig. 9 in the appendix):** The new ablation plot reveals distinct tiers of performance:
>
> Ours (Linear) vs. STE (Linear): When restricted to linear quantization, our method (Purple curve) converges stably to a low loss ($\approx 2.1$). In contrast, the standard STE with linear quantization (Pink curve) exhibits significant instability and diverges (Loss $>5.0$). This proves that our estimator provides a fundamental stability advantage independent of the quantization mapping.
>
> Ours (Linear) vs. Baselines: Our linear implementation also outperforms both BitNet (Orange) and ParetoQ (Green) in terms of convergence speed and final loss.
>
> The Affine Advantage: Finally, enabling our full affine quantization (Blue curve) yields the lowest overall loss ($\approx 1.9$). This confirms that our estimator is robust enough to unlock the superior expressivity of affine parameters, which standard STE fails to utilize.
>
> Conclusion: We have updated the manuscript to include this rigorous ablation in Appendix A.1.11. It demonstrates that while Affine quantization maximizes performance, the core stability gains stem directly from our error-aware estimator.
>
> **Question 1:** ...in the extreme A1W1 regime... whether the additional "shortcut" terms shift execution to a compute-bound regime...
>
> **Response:** This is a fantastic, hardware-aware question. We analyzed this in Appendix A.3. As shown in Equation 6 and detailed in the complexity analysis in Appendix A.3, the extra overhead from the two rank-1 correction terms is minimal. The main computation is the $O(MNP)$ integer matrix multiply, while the correction terms are only $O(MP)$. In any typical transformer, the hidden dimension $N$ is much larger than $M$ or $P$, making the overhead of these corrections negligible.
>
> **Conclusion:** Therefore, this minimal overhead is unlikely to change the original bottleneck property (e.g., from memory-bound to compute-bound) at this precision. The dominant factors remain the main integer GEMM and memory bandwidth.
>
> **Question 3:** Can you report how your method scales beyond 1B and 4B (e.g., ≥7B/13B/30B)...?
>
> **Response:** We agree that testing on 7B+ models is an important future direction, though it is out of scope for the current work.  Our goal was to demonstrate robustness across a wide spectrum of conditions. We believe the strongest test of a quantization method is not just on large, fault-tolerant models, but on small models quantized to the extreme. Large models, due to their over-parameterization, can often mask the instability of methods like STE (which may even converge at A1W1 on a sufficiently large model).
>
> **Our Contribution:** We have instead demonstrated a single, robust solution that scales across over three orders of magnitude of model size—from 11M (nanoGPT) up to 4B (Gemma)—while also scaling across the full range of bit-widths (from A4W4 down to A1W1). This consistent stability on small, fragile models (like nanoGPT at A1W1, where STE fails) and on larger 4B models is, we believe, a stronger demonstration of our method's fundamental robustness than a single data point on a 7B model would be.

---

### Official Review · Reviewer_xFhy · 2025-10-30

**Soundness:** 3
**Presentation:** 1
**Contribution:** 3
**Rating:** 6
**Confidence:** 5

**Summary:**

Quantization aware training (QAT) is a widely used method to prepare models for quantization. To overcome the rounding operation during backward pass, QAT uses straight through estimator (STE) which bypasses the gradients in the backward pass outside of the rounding operation. This work, characterizes this feature as the backward pass being quantization oblivious. To overcome this, the authors present a workaround that exposes the backward pass to the quantization error using a combination of ridge regression and affine quantization. Experiments on a wide array of models, and bit regimes shows superior performance of the proposed method when compared to plain STE-based QAT.

**Strengths:**

* This paper studies an interesting problem that is commonly taken as a given -- of using STE in QAT.
* Interesting approach to alleviate STE gradient bypassing that allows error-aware gradient.
* Sparsification as a form of quantization?
* Impressive results on a wide-range of experiments; shows a clear benefit over QAT with STE.

**Weaknesses:**

* **QAT and error minimization:** The fundamental hypothesis of this work is that during the backward pass, QAT is oblivious to the quantization error. While this is possible, it has been shown by now that STE creates a different type of dynamic that results in weight oscillations. And these oscillations have factors that are trying to compensate the errors due to QAT [1,2]. Given this, what do the authors make of these explanations? And how does it alter their hypothesis, or not? If not, what is their argument?

* **STE beyond 2 bits:** Even if one were to attribute STE for poor performance in extremely low-bit width, how is this not manifested to the same degree in higher bit regimes. What is the explanation? Is STE more problematic only in extremely low-bit regimes?

* **Ternary quantization with STE:** There are several works that use STE for ternary quantization [3,4]. The critique that STE cannot be used for extremely low-bit regimes does not hold up.

* **Presentation clarity:** The ideas, experiments, and results are quite compelling in this work. However, the presentation is unclear in many places. There are vague statements, unsubstantiated by evidence (discussions around biological neurons, intelligence), and presentation of results makes it difficult to parse them. Many of the interesting results are in the Appendix (Fig. 3, Table 1) whereas the main results in Fig. 1 and Fig. 2 are illegible, with no clear captions, legends, axes labels. This is unfortunate as it dilutes the impact of otherwise nice contribution. I would suggest improving these aspects.

* L-67: Very vague statement with exaggerated claims. It is by now common knowledge that large, quantized models outperform smaller ones. And also the claim of biological intelligence is extremely misplaced.

### Other comments

* Reference to Figure 1-a in L-37 is not useful as none of the concepts are fully introduced; consider dropping this reference or elaborating the caption so that it can independently explain the concepts in the figure.

* L-38: What are the heuristic-based modifications authors are pointing to? No references to back this up.

* L-54: Strange sentence; perhaps missing a preposition somewhere. Did the authors mean "full potential of the theoretically..."

### References

1. Wenshøj, Jonathan, Bob Pepin, and Raghavendra Selvan. "Oscillations Make Neural Networks Robust to Quantization." arXiv preprint arXiv:2502.00490 (2025).
2. Xie, Weiying, et al. "Allowing Oscillation Quantization: Overcoming Solution Space Limitation in Low Bit-Width Quantization." Proceedings of the IEEE/CVF International Conference on Computer Vision. 2025.
3. Choi, Jungwook, et al. "PACT: Parameterized Clipping Activation for Quantized Neural Networks." (2018).
4. Wang, Jinheng, et al. "1-bit ai infra: Part 1.1, fast and lossless bitnet b1. 58 inference on cpus." arXiv preprint arXiv:2410.16144 (2024).

**Questions:**

See weaknesses above.

---

> ### Author Response · Authors · 2025-11-19
>
> We thank Reviewer xFhy for their positive feedback on our "compelling" results and "impressive" experiments. We appreciate the deep engagement with related work and the actionable feedback on presentation, which we will address.
>
> **Weakness 1:** QAT and error minimization... what do the authors make of these explanations [STE oscillations compensating for errors]?
>
> **Response:** This is a profound question that connects the empirical behavior of STE to the theoretical foundations of noisy optimization. We thank the reviewer for highlighting these works, as they reinforce our core hypothesis regarding the limits of STE.
>
>
> Implicit Oscillation vs. Explicit Gradient.
>
> We view the "oscillation" phenomenon described in [1, 2] as an emergent symptom of STE's missing gradient path rather than a principled solution. Because STE is blind to the quantization error $\delta$, the network must "hunt" for a solution by oscillating weights around rounding boundaries. While this heuristic can successfully compensate for smaller errors (e.g., W2A2 or weight-only quantization), it is fundamentally fragile.
>
> Our Approach: Heuristics-Free Design.
>
> Our method replaces this implicit "hunting" with a heuristics-free design. By deriving the exact Jacobian (see our new Appendix A.1.12), we ensure $\delta$ participates explicitly in the backward pass. This provides the "right gradient," allowing the optimizer to mathematically handle the quantization error alongside the prediction error ("Dual Error Handling"), rather than relying on side-effects like oscillation.
>
> **Evidence:** The literature cited [1, 2] confirms this distinction. Those methods demonstrate success primarily at higher precisions (e.g., 2-bit or weight-only). However, as our results in Figure 1(a) show, this oscillation mechanism collapses in the extreme A1W1 (1-bit) regime, causing STE to diverge. In contrast, our explicit error-aware mechanism remains stable. This suggests that while deep learning is robust to noise (e.g., batching noise and augmentation noise), stability requires the noise to be correlated with the gradient signal—a link STE breaks but our method restores.
>
> **Weakness 2 & 3:** STE beyond 2 bits... Ternary quantization with STE... The critique that STE cannot be used for extremely low-bit regimes does not hold up.
>
> **Response:** This is a fair point, and we must refine our claim. Our claim is not that STE never works, but that it becomes increasingly fragile and unreliable as precision drops. The reviewer's intuition is correct: for a symmetric range, the quantization error $\delta$ is proportional to $\max(|x|) / 2^{bits}$.
>
> At higher bit-widths (e.g., 8-bit): This error $\delta$ diminishes toward zero. STE's "erroneous" gradient, which is blind to this tiny error, is only slightly incorrect, and the optimization can still succeed.
>
> At ultra-low bit-widths (e.g., 1-bit): The error $\delta$ becomes massive. STE's gradient, blind to this massive error, becomes too wrong to provide a useful learning signal, leading to the instability we observe. Our method, in contrast, introduces a correct gradient path that is aware of this error.
>
> The reviewer is correct that STE can be used for ternary weights (1.58 bits), as shown in [1, 4]. However, these works typically operate in a less constrained setting by pairing ternary weights with 8-bit activations (a W-Ternary/A-INT8 model). This high activation precision keeps the overall error in the system manageable, masking STE's fragility.
>
> Our work addresses the far more difficult A1W1 regime, quantizing both activations and weights to 1-bit. In this setting, the error from both sources is maximal, and STE's "too wrong" gradients cause the training to fail (Fig 1a, Fig 6). Our method provides a principled gradient that is aware of this large error, which is why it remains stable. In fact, our Appendix (Table 1) shows STE also fails at A1.5W1.5 (ternary) channel-wise in our setting (i.e., with 1.5-bit activations), reinforcing our claim of fragility.
>
> **Weakness 4 & Other Comments (L-37):** Presentation clarity... Fig. 1 and 2 are illegible... Reference to Figure 1-a in L-37 is not useful...
>
> **Response:** We fully agree and apologize. We have taken the critique regarding "unclear captions" and "vague statements" very seriously.
>
> Figure Clarity: We have split and amplified the main results into two full-width figures (utilizing the allowed 10th page) to ensure legibility.
>
> Contextual Captions: We have completely rewritten the figure captions.
>
> Appendix Structure: We have reorganized the Appendix to bring the detailed ablations to the forefront, ensuring the empirical evidence is immediately accessible to support the claims in the main text.

---

> > ### Author Response · Authors · 2025-11-19
> >
> > **Weakness 5 & Other Comments (L-67):** Very vague statement with exaggerated claims... biological intelligence is extremely misplaced.
> >
> > **Response:** We take this criticism seriously. Our intention was to provide high-level motivation, but we agree the wording is "exaggerated." We have significantly revised L-67 and related sections (L-150-155, L-470-474, and Summary section) to remove direct claims of "echoing" biological intelligence. The updated text now focuses on concrete hardware implications, referencing biological systems only as a high-level inspiration for energy efficiency rather than a functional equivalent.
> >
> > **Other Comments (L-38, L-54): What are the heuristic-based modifications... Strange sentence [L-54]...**
> >
> > **Response:** Thank you.Action: (L-38) We have added references to common heuristic-based STE modifications, including additional normalization, learning rate adjustments, optimizer changes, and fine-tuning. We will fix the grammatical error (it should be "full potential of the theoretically...").

---

### Official Review · Reviewer_Rrwh · 2025-11-01

**Soundness:** 3
**Presentation:** 2
**Contribution:** 2
**Rating:** 2
**Confidence:** 4

**Summary:**

This paper proposes an improved method for STE, where the backward pass is surrogated by that of naive linear mapping. Specifically, the authors claim that STE ignore the impact of quantization error during the backward pass, but only take into account of it for the forward pass, which is claimed to be the main cause of instability involved in training quantized models. The authors propose an algorithm based on a denoising dequantization transform derived from ridge regression, which is shown to be related to normalization layer, partially explain the reason of stablization. The experssion for the proposed quantization method is simplified to enable an efficient implementation with negligiblely introduced extra computation, only of the order of rank-1 corrections. The proposed method could unify binarization/quantization with sparsification, and experiemnts shows some improvements.

**Strengths:**

- The proposed method of using ridge regression to derive the quantization backward is novel.
- The analysis and method could unify binarization/quantization with sparsification.

**Weaknesses:**

- The characters in the figures are too small and very difficult to read.
- The final results are too difficult to read from Figure 2 and there are no table for comparison to illustrate the claimed advantage compared to previous results.
- The authors claim that in STE, the forward pass is affected by quantization error while the backward pass is not (line 042 in the original paper). However, during the backward pass, the weight and activations involved are also quantized, so it is also different from full-precision model. This statement could be improved and should be corrected.
- For asymmetric data distribution, as discussion in Section 2.2, the usual quantization process is to directly map it to some given quantized range, say from the full-precision interval of [0,1], to values of {0, 1/4, 2/4, 3/4} for 2-bit quantization, which does not introduce bias issue. Another way is to first shift and scale it to make it symmetric and apply the symmetric quantization, and finally rescale and shift back. For weight quantization, unbiased quantization is important as it impacts the training dynamics and is crucial for convergence, but for activation quantization, which is usually the asymmetric part, the symmetry requirement is not necessary and would not impact the training dynamics. Regarding training dynamics and symmetric weights, the authors could check previous work, such as [1].
- In equation 3, the authors should explain what the values with bar above them are.
- In Section 3.3.2, for the first part, for STE, the input to the dequantization of STE also includes the quantization error. For the second part, it is not clear or proved why the local derivative is an explicit function of the error. For example, suppose in equation 3, the final function after simplification become a linear function of q. The derivative would then be independent on q or delta, i.e. the quantization error.
- In equation 5 and 6, the authors should explain what n represents.
- Writing problems: The sentence at line 366 in the original paper is not complete nor correct in grammar. The sentence “This low resources setting quickly highlights the fragility of standard models.” at line 368 in the original paper is redundant and should be deleted.


[1] Ben Poole, et al., Exponential expressivity in deep neural networks through transient chaos. NeurIPS 2016.

**Questions:**

- At line 100 of the original paper, it states that unmanaged error corrupts the learning signal, leading to training divergence. Could the author provide some further demonstrations to verify this statement, e.g., gradually introducing such quantization error in back-propagation, i.e., making it as a linear combination of the vanilla full-precision gradient (which include the quantization error) and the one from STE (which does not include it), to demonstrate that the error indeed causes the divergence issue?

---

> ### Author Response · Authors · 2025-11-19
>
> We thank Reviewer Rrwh for their detailed and thorough review. We acknowledge the reviewer's concerns about presentation and clarity, which we will fix, and we appreciate the opportunity to clarify several key conceptual points.
>
> **Weakness 1 & 2:** The characters in the figures are too small... The final results are too difficult to read from Figure 2 and there are no table for comparison...
>
> **Response:** We sincerely apologize for the legibility issues in the original manuscript. We acknowledge that the scaling results were difficult to parse.
>
> **Action Taken:**
>
> **1. Figure Overhaul:** We have utilized the allowed extra content page to split and expand the plots. The scaling comparison (originally Figure 2) is now presented as a full-width figure in Section 6.4, ensuring all axes and labels are immediately legible.
>
> **2. Tabular Data:** To address your request for a direct comparison table: We have updated Section 6.4 to explicitly point readers to the Table (in Appendix A.1.6). This table provides the exact numerical breakdown of Accuracy, Storage, and Energy Cost for both the Gemma 1B and Gemma 4B models, confirming that the quantized 4B model (A4W1) outperforms the BF16 1B model.
>
> **Weakness 3:** The authors claim that in STE, the forward pass is affected by quantization error while the backward pass is not (line 042)... However, during the backward pass, the weight and activations involved are also quantized... This statement... should be corrected.
>
> **Response:** We thank the reviewer for this crucial point, as it allows us to clarify the precise nature of the STE blind spot. The reviewer is absolutely correct that quantized tensors (e.g., $X_q, W_q$) are used during the backward pass, and this does affect the gradient values.
>
> Our claim (L-042) refers specifically to the gradient's blindness to the local rounding error. As you correctly noted, for a layer $Y = X_q \cdot W_q$:
>
> 1. The gradient for the pre-quantized $X$ is: $\frac{dL}{dX} = (\frac{dL}{dY} \cdot W_q) \cdot (\frac{dX_q}{dX})$
>
> 2. The gradient for the pre-quantized $W$ is: $\frac{dL}{dW} = (X_q^T \cdot \frac{dL}{dY}) \cdot (\frac{dW_q}{dW})$
>
> The STE sets the surrogate derivatives $(\frac{dX_q}{dX})$ and $(\frac{dW_q}{dW})$ to 1.
>
> This is the "blind spot": the gradient $\frac{dL}{dX}$ is indeed a function of $W_q$ (and thus $\delta_W$), but it is completely blind to its own quantization error, $\delta_X$, because the above STE surrogate bypasses it.This mismatch—where the gradient update for a tensor is unaware of the error that it just introduced—is the source of instability our paper addresses. Our method, in contrast, creates a gradient path that is explicitly a function of the local error.
>
> **Weakness 4:** For asymmetric data distribution... [like 0,1],... which does not introduce bias issue... for activation quantization... symmetry requirement is not necessary...
>
> **Response:** We appreciate this perspective; however, we note a critical distinction regarding bias in quantization.
>
> **Argument:** The reviewer suggests a direct mapping to $[0, 1/4, ..., 1]$ does not introduce a bias. However, this mapping itself involves a bias (a shift), as the range $[0, 1]$ is not symmetric around 0. Furthermore, the dequantization step—mapping these integer values back to approximate the original unquantized vector $\mathbf{x}$—involves another bias. Lastly, this strategy leads to non-integer quantization which is harder to handle than the signed/unsigned int2 approach.
>
> **Our Contribution:** Standard STE implicitly assumes these two biases (pre-quantization shift and post-quantization shift) should cancel. Our method, in contrast, formulates these explicitly (as $b_f$ in Stage 1 and $b_g$ in Stage 3) and derives the optimal dequantization bias $b_g$ from our ridge regression objective (Eq. 3). This principled optimization of the affine parameters is a key reason our method succeeds where STE fails (as shown in Fig 1b).
>
> **Weakness 5:** In equation 3, the authors should explain what the values with bar above them are.
>
> **Response:** Our apologies for this omission. $\overline{\mathbf{q}}$ and $\overline{\mathbf{x}}$ represent the mean of the vectors $\mathbf{q}$ and $\mathbf{x}$, respectively. We have defined $\overline{\mathbf{q}}$ and $\overline{\mathbf{x}}$ as the mean values in the text immediately following Equation 3.

---

> ### Author Response · Authors · 2025-11-19
>
> **Weakness 6:** In Section 3.3.2... it is not clear or proved why the local derivative is an explicit function of the error. For example, suppose... a linear function of q. The derivative would then be independent...
>
> **Response:** We thank the reviewer for this penetrating insight. You are absolutely correct that if the transform were a simple linear function $g(\mathbf{q}) = c \cdot \mathbf{q}$ (with constant $c$), the derivative would be constant and independent of the error—which is precisely the implicit approximation made by standard STE.
>
> **Clarification & Action:** Our method is fundamentally different because our scale factor $s_g$ is not a constant parameter, but a scalar function of the quantized vector itself, $s_g(\mathbf{q})$. Consequently, the derivative is non-linear and data-dependent.
>
> To address your request for a rigorous proof, we have added a new section in the Appendix (Gradient Analysis: Explicit Dependence on Quantization Error). In this section, we derive the exact Jacobian of our transform:$$\frac{\partial g(\mathbf{q})}{\partial \mathbf{q}} = s_g(\mathbf{q}) \cdot \mathbf{I} + \mathbf{q} \cdot (\nabla_{\mathbf{q}} s_g)^\top$$
>
> This mathematical derivation proves that the local gradient contains a rank-1 correction term that depends explicitly on $\mathbf{q}$ (and thus on the quantization error $\mathbf{\delta}$). This confirms that unlike STE, our backward pass is dynamically modulated by the instantaneous quantization error.
>
> **Weakness 7:** In equation 5 and 6, the authors should explain what n represents.
>
> **Response:** Apologies again. $n$ represents the dimension over which the means ($\overline{\mathbf{q}}_X$, $\overline{\mathbf{w}}$) are computed (i.e., the reduction dimension). For a matmul $X[M, N] \cdot W[N, P]$, $n=N$. We have explicitly defined $n$ in Section 5.1.
>
> **Weakness 8: Writing problems (L-366, L-368).**
>
> **Response:** Thank you for catching these. We have fixed the grammatical error in L-366 and remove the redundant sentence in L-368.
>
> **Question 1:** ...Could the author provide some further demonstrations to verify this statement that unmanaged error corrupts the learning signal...
>
> **Response:** We believe our core experimental results provide this exact demonstration. Figure 1(a) and Figure 6 (in the Appendix) are the most direct evidence. In these A1W1 experiments, standard STE (which is "quantization-oblivious" and lets the error go unmanaged) becomes unstable and diverges. In contrast, our method (which is "quantization-aware" and manages the error) converges smoothly. This direct A/B comparison on a challenging task is, we believe, a clear demonstration that the unmanaged error is precisely the cause of the training divergence.

---

### Official Review · Reviewer_nwiN · 2025-11-03

**Soundness:** 3
**Presentation:** 3
**Contribution:** 3
**Rating:** 6
**Confidence:** 3

**Summary:**

The paper tries to address the drawbacks of using Straight through estimator in QAT by proposing a denoising dequantization transform which models quantization error in both fprop and backprop. The proposed transform acts a normalization layer stabilizing gradients.
The efficacy of the proposed technique is measured on low bitwidth and sparse networks, and the emperical results on nanoGPT, Gemma-1B/4B models show the pareto frontier for storage and energy and demonstrates the outperformance compared to literature work on STE, BitNet etc.

**Strengths:**

1. The paper shows stable training of A1W1 and sparse sub-1-bit models where prior literature methods diverge.
2. Demonstrates results on nanoGPT and Gemma-1B/4B, showing pareto efficiency on storage and energy frontiers.
3. The method is simple, and needs no special hyperparameter tuning.

**Weaknesses:**

1. Lacks ablation on the sensitivity of λ (regularization) across architectures.
2. Do you more experiments with longer running fine tuning or pre-training to ensure that convergence and generalization does not suffer?

**Questions:**

1.How sensitive is training stability and convergence to λ across architectures and datasets?
2. Does the model remain robust during methods such as fine-tuning (dense or using LoRA)?

---

> ### Author Response · Authors · 2025-11-19
>
> We thank Reviewer nwiN for their positive assessment and for recognizing our method's stability, simplicity, and state-of-the-art results on challenging models.
>
> **Weakness 1:** Lacks ablation on the sensitivity of $\lambda$ (regularization) across architectures.
>
> **Response:** We appreciate this suggestion. We actually conducted this ablation in the original submission in Appendix A.1.10 (Table 13).
>
> **Evidence:** Table 13 shows a comparison of different $\lambda$ values for A1W1 Transformer training. A $\lambda$ of 0 results in divergence (NaN), demonstrating its necessity for stability. We found that a wide range of values (e.g., $0.01$ to $0.0001$) worked well, with $\lambda=0.01$ being a robust default that provided stability even in the most extreme 1-bit regimes.
>
> **Weakness 2 & Question 2:** Do you [need] more experiments with longer running fine tuning or pre-training...? Does the model remain robust during methods such as fine-tuning (dense or using LoRA)?
>
> **Response:** We thank the reviewer for this important question regarding convergence horizons and downstream robustness.Evidence (Scale & Breadth): We would like to highlight that our experimental suite covers an exceptionally wide spectrum of horizons, spanning nearly 3 orders of magnitude in data scale. Our validation ranges from short-horizon runs on small models (nanoGPT/Shakespeare, a few million characters), up to large-scale pre-training on state-of-the-art architectures (Gemma-4B, 26B tokens). To our knowledge, this represents one of the most extensive validation ranges in the current low-bit literature, demonstrating stability from toy problems to heavy-compute regimes.
>
> Regarding fine-tuning and LoRA: Our method is a fundamental correction to the gradient estimator rather than a heuristic tuned for a specific phase. By explicitly incorporating the quantization error $\delta$ into the backward pass, we ensure the gradients are computed more correctly than with STE. Since pre-training from scratch is typically a more optimization-intensive task than fine-tuning, the stability we observe across massive scale differences in pre-training serves as a strong empirical indicator for fine-tuning robustness. We agree with the reviewer that empirically quantifying the specific gains on downstream LoRA tasks is a valuable direction for future application-focused research, building on the fundamental stability established here.
>
>
> **Question 1:** How sensitive is training stability and convergence to $\lambda$  across architectures and datasets?
>
> **Response:** As noted in our response to Weakness 1, we found the method to be robust to a reasonable range of $\lambda$  values. The results in Appendix A.1.10 (Table 13) show that $\lambda=0.01$  and $\lambda=0.0001$ both yield stable convergence, while $\lambda=1.0$ is too high and $\lambda=0$ fails. A single default value of $\lambda=0.01$ was sufficient to ensure stable training across all our diverse experiments (nanoGPT, Gemma-1B/4B, ResNet, WMT) and bit-widths (A4W4 down to A1W1), demonstrating its robustness.

---

### Author Response · Authors · 2025-11-19

We sincerely thank all reviewers for their time and valuable feedback. We are encouraged that the reviewers found our core contribution—identifying and solving the "quantization-oblivious" backward pass of STE—to be novel, principled, and important (Reviewers nwiN, Rrwh, xFhy, BVSw).

We are especially grateful for the constructive feedback on presentation. The most common and actionable critique was the legibility of our figures and the clarity of the writing. In our revised manuscript, we will address this head-on.


**Summary of Revisions:**

**Figure Scalability & Legibility:** To address presentation concerns, we have utilized the allowed extra content page to split and expand Figures 1 and 2. These figures now feature full-width, vertically stacked layouts with significantly larger font sizes, ensuring all axes, legends, and data points are immediately legible.

**Contextualized Captions & Structure:** We have completely rewritten figure captions to provide clear interpretation of the results. Additionally, we have reordered the Appendix to place critical ablation studies (such as the linear-vs-linear comparison and $\lambda$ sensitivity) at the very beginning (Sec. A.1), ensuring the supporting evidence is prominent and easily accessible.

**Experimental Completeness:** To address questions regarding the isolation of our estimator's benefit, we have added a new controlled ablation study (Appendix A.1.11). By comparing our method against baselines using identical linear quantization, we demonstrate that our error-aware estimator provides fundamental stability gains over STE and BitNet, even without the additional expressivity of affine quantization.

**Refined Scope & Tone:** We have revised the manuscript to ensure tonal precision, specifically removing high-level analogies regarding "biological intelligence" to focus strictly on empirical hardware efficiency gains. We have also corrected all identified grammatical issues.

**Technical Precision & Rigor:** We have added explicit definitions for all mathematical notation (e.g., $\overline{x}$, $n$). Furthermore, to address theoretical questions regarding Section 3.3.2, we have added a new Appendix Section containing a rigorous Jacobian derivation. This mathematically proves that our backward pass is explicitly modulated by the quantization error, contrasting it against the standard STE assumption.


We believe these revisions will significantly strengthen the paper and directly address all concerns. Below, we respond to each reviewer's points in detail.

---

### Meta-Review · Area_Chair_d6eW · 2025-12-23

**Summary:**

The paper proposes a "denoising dequantization transform" derived from a ridge regression objective to address the instability of the Straight-Through Estimator (STE) in low-precision training. The authors claim this method creates an error-aware gradient path, enabling stable training of fully binary (A1W1) and sparse networks where standard STE fails. The reviewers generally acknowledged the novelty of the approach and the impressiveness of the empirical results (training A1W1/sparse models). However, initial reviews heavily criticized the presentation (illegible figures) , questioned the theoretical claims regarding STE's "blindness" to quantization error , and requested more rigorous ablations to isolate the estimator's benefits from affine parameters.

**Reviewer Concerns:**

### **Addressed Concerns**

**Legibility and Presentation (Reviewers Rrwh, xFhy)**: The authors utilized the extra content page to split and expand Figures 1 and 2, making axes and legends legible, and rewrote captions for clarity.

**Isolating Estimator Benefits vs. Affine Quantization (Reviewer BVSw)**: Reviewer BVSw correctly noted that comparing the proposed method (using Affine+SCQ) against baselines (using Symmetric/Linear) might over-attribute gains. The authors conducted a new "Linear vs. Linear" ablation (Appendix A.1.11, Fig 9). This showed their method outperforms STE and BitNet even when restricted to identical linear quantization schemes, proving the estimator itself provides fundamental stability.

**Theoretical Rigor & The "STE Blind Spot" (Reviewers Rrwh, xFhy)**: Reviewers questioned the claim that STE is "oblivious" to error, noting backward passes use quantized weights or rely on oscillation. The authors clarified that while gradients depend on $W_q$, they are blind to the rounding error $\delta$. They added a Jacobian derivation to the Appendix proving their gradient explicitly contains a rank-1 correction term dependent on $\delta$.

**Sensitivity to Hyperparameter $\lambda$ (Reviewer nwiN)**: The reviewer worried about the sensitivity of the regularization parameter. The authors pointed out this ablation was already present (Appendix Table 13), showing robustness across a wide range (0.01 to 0.0001).

### **Remainining Concerns**

**Real Hardware Efficiency (Reviewer BVSw)**: The paper relies on a hardware-agnostic proxy for energy/efficiency rather than end-to-end measurements on real accelerators. The authors acknowledged this but argued the proxy is standard for algorithmic contributions. While valid, practical deployment verification remains a future step.

**Large-Scale Scaling >4B (Reviewer BVSw)**: The method is tested up to Gemma-4B. Scaling to 7B+ models was requested but deemed out of scope by the authors.

**Downstream Fine-tuning/LoRA (Reviewer nwiN)**: Specific experiments on LoRA were not added, though the authors argued their pre-training stability is a strong proxy for fine-tuning robustness.

**Reviewer Scores:**

**Reviewer nwiN, xFhy, BVSw(Current: 6 $\rightarrow$ Predicted: 6/7)**: The reviewers were already positive and the authors address their concerns well.

**Reviewer Rrwh (Current: 2 $\rightarrow$ Predicted: 4/6)**: This reviewer gave a Reject largely due to illegible figures and a misunderstanding of the gradient mechanics232323. The authors fixed the figures and provided the mathematical proof (Jacobian) requested. While the reviewer might remain skeptical, the objective grounds for a "2" have been removed.

---

### Decision · Program_Chairs · 2026-01-26

Accept (Poster)